# Deep Black-Box Reinforcement Learning with Movement Primitives

**Fabian Otto**[1,2]**, Onur Celik**[3]**, Hongyi Zhou**[3]**, Hanna Ziesche**[1]**, Ngo Anh Vien**[1]**,**
and **Gerhard Neumann**[3]

[1] Bosch Center for Artificial Intelligence, Germany
[2] University of Tübingen, Germany
[3] Autonomous Learning Robots, Karlsruhe Institute of Technology, Germany
`fabian.otto@bosch.com`

**Abstract:** *Episode-based reinforcement learning* (ERL) algorithms treat *reinforcement learning* (RL) as a black-box optimization problem where we learn to select a parameter vector of a controller, often represented as a movement primitive, for a given task descriptor called a context. ERL offers several distinct benefits in comparison to step-based RL. It generates smooth control trajectories, can handle non-Markovian reward definitions, and the resulting exploration in parameter space is well suited for solving sparse reward settings. Yet, the high dimensionality of the movement primitive parameters has so far hampered the effective use of deep RL methods. In this paper, we present a new algorithm for deep ERL. It is based on differentiable trust region layers, a successful on-policy deep RL algorithm. These layers allow us to specify trust regions for the policy update that are solved exactly for each state using convex optimization, which enables policies learning with the high precision required for the ERL. We compare our ERL algorithm to state-of-the-art step-based algorithms in many complex simulated robotic control tasks. In doing so, we investigate different reward formulations - dense, sparse, and non-Markovian. While step-based algorithms perform well only on dense rewards, ERL performs favorably on sparse and non-Markovian rewards. Moreover, our results show that the sparse and the non-Markovian rewards are also often better suited to define the desired behavior, allowing us to obtain considerably higher quality policies compared to step-based RL.

**Keywords:** Movement Primitives, Deep Episode-Based RL, Trust Region Layers

## 1 Introduction

*Reinforcement learning* (RL) problems can be viewed from a step-based [1, 2, 3] and an episode-based perspective [4, 5, 6]. In the former, most commonly found in deep RL, a policy selects an action for each state of the trajectory. Consequently, step-based RL requires Markovian reward definitions. Moreover, the exploration in action space often induces a random walk that inadequately explores the entire behavior space of the agent. As a result, most step-based RL algorithms only work well with dense rewards, where the agent receives a reward signal at each time step, rather than only at the last step. We refer to the latter as a sparse reward setting.

In the *episode-based reinforcement learning* (ERL) perspective, we choose the complete behavior during the episode with respect to the controller parameters in the beginning of the episode. Typically, simple controllers are used that are valid only for executing a single trajectory, such as *movement primitives* (MPs). At the beginning of an episode, the MP parameters are adapted to a context vector that serves as task descriptor and may contain e.g. different initial joint configurations, the goal to reach, or start positions as well as object and obstacles locations. ERL allows for efficient exploration of the behavior space because exploration is implemented in the parameter space of the MP, allowing these algorithms to learn from sparse or even non-Markovian rewards. In addition, ERL inherently generates smooth control trajectories due to the use of MPs, which often leads to more energy efficient behavior. However, ERL has so far not gained much popularity as

6th Conference on Robot Learning (CoRL 2022), Auckland, New Zealand.

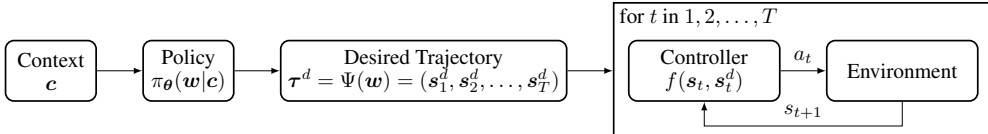

Figure 1: Overview of the proposed Black-Box Reinforcement Learning (BBRL) framework. The normal distributed policy predicts, given the context $c$, the parameters of a movement primitive that translates to a desired trajectory $\tau^d$. A trajectory tracking controller $f$ generates low-level actions $a_t$ given the current $s_t$ and the desired state $s_t^d$ from the trajectory $\tau^d$.

it does not exploit the time-series structure of the RL problem and is therefore considered less data efficient, at least in the dense reward setting. So far, there is no deep learning method to adapt the MP parameters to the context, which limits ERL to linear adaptation strategies [7, 8].

This paper offers two main contributions: (i) We propose a new algorithm for deep ERL. The high-dimensional action space imposed by the MP parameters requires highly accurate action selection, hence robust and precise policy updates are needed. We adapt a recent step-based RL algorithm based on differentiable *trust region projection layers* (TRPLs) [9] to the ERL setting. Unlike methods such as *proximal policy optimization* (PPO) [2], that are based on approximated trust regions, TRPL offers exact trust regions for the policy updates which is essential for learning precise policies in high-dimensional action spaces. As second contribution, (ii) we show that ERL can be superior to step-based RL for tasks that are more easily defined by sparse or non-Markovian rewards than by a dense reward. To this end, we have investigated different reward formulations for a variety of complex simulated robot control tasks. Our experiments show that step-based algorithms have difficulties learning high-quality policies in sparse or non-Markovian reward settings, while our proposed algorithm scales well to these reward settings. We further show that dense Markovian rewards can be very restrictive in terms of specifying a desired behavior in our domains. For example, if we want to achieve energy efficient, time-optimal reaching motions, sparse rewards can be used that only penalize the distance to the goal in the last time step. Non-Markovian rewards are useful for tasks with inherent non-Markovian optimality descriptors, e. g. , the maximum height during a jump or the minimum distance of a bat to a ball. As a consequence, our algorithm can master tasks where step-based RL performs poorly even after intensive engineering of the used dense rewards.

In summary, our findings show that our algorithm has the following benefits: (a) it produces inherently smooth control policies, (b) offers efficient exploration in parameter space, which in turn enables learning with (c) sparse and non-Markovian rewards. These rewards also (d) allow for a more direct specification of the desired behavior, which leads to (e) higher quality of the learned behavior. These benefits come at the cost of an increased data complexity (a factor of 2 to 3).

## 2  Background and Related Work

**Black-Box Reinforcement Learning.**  Contextual episode-based policy search [4, 8] treats RL as black-box optimization problem where a contextual search distribution $\pi(w|c)$ over the controller parameters $w$ is optimized to maximize the expected return $R(w, c)$, i. e. ,

$$\arg\max_{\pi(w|c)} \mathbb{E}_{p(c)} \left[ \mathbb{E}_{\pi(w|c)}[R(w, c)] \right] ,$$

where $p(c)$ denotes the context distribution given by the task. Due to the black-box nature of the problem, no structural assumptions are made for the return function $R(w, c)$, i. e. , the return can be any non-Markovian function of the resulting trajectory. Most ERL algorithms only consider the non-contextual setting, where different optimization techniques have been used, such as policy gradients [10], natural gradients [11], stochastic search strategies [12, 13, 14], or trust-region optimization techniques [5, 6, 8]. The only methods that incorporate context adaptation [8, 14] consider a linear mapping from context to parameter space. We, in contrast, consider highly nonlinear context-parameter relationships using deep learning.

As alternative to gradient- and step-based methods recent works [15, 16, 17] also proposed using a full gradient-free black-box approach for finding the optimal neural network parameters. They typically consider learning the neural network policy parameters as the black-box optimization problem

and leverage the episode performance for evaluating network configurations. While these methods can be competitive for some tasks, they are still difficult to use for most contextual setups as they require rollouts with different neural network parameters to be comparable. Yet, in the contextual case the performance of the rollout does not only depend on the neural network parameters but also on the context (e. g. the goal) which makes the evaluation much more noisy. For example, a good neural network parametrization could still yield a poor performance because it has been evaluated in a hard context. The above approaches are completely ignorant to the context and, therefore, cannot attribute the performance differences to the context.

In contrast, our setting does not perform black-box optimization on the level of a global neural network control policy with several thousand parameters but on the level of local control parameters (in the range of 20-50 dimensions). While we, unlike the above linear approaches, still use a neural network policy with a high number of parameters to adapt the local control parameters to the context, parameters are *not* treated as a black-box. Instead, the DNN policy is updated using policy gradient, which utilizes the context information as well as the derivatives of the policy.

**Step-Based Reinforcement Learning.** Unlike episode-based methods, step-based approaches maximize the expected return by optimizing an action-selection policy that chooses a new action in each time step of the trajectory. While the goal remains to learn optimal trajectories, they interact directly with the environment based on raw actions, such as positions, velocities, or torques generated from the current state information. One such method is the trust region policy optimization algorithm [18], which first introduced trust region methods to deep RL, but is rather complex and difficult to scale. PPO [2] simplifies the update by introducing a clipping heuristic to the policy gradient objective. An alternative to the above policy gradient methods are policy iteration based approaches [19].

**Differentiable Trust Region Layer.** Differentiable *trust region projection layers* (TRPLs) [9] present a scalable and mathematically sound approach for enforcing trust regions in step-based deep RL. While PPO [2] is also motivated by trust-region updates of the policy, it cannot enforce the trust region exactly. TRPL provides an efficient way to enforce a trust region as well as more stability and control during training, while reducing the dependency on code level choices [20].

The layer takes the output of a standard Gaussian policy as input in terms of mean $\boldsymbol{\mu}$ and variance $\boldsymbol{\Sigma}$ and projects it into the trust region if the given mean and variance violate their respective bounds. This projection is done for each input state individually. Subsequently, the projected Gaussian policy distribution with parameters $\tilde{\boldsymbol{\mu}}, \tilde{\boldsymbol{\Sigma}}$ is used for any further steps, e. g. for sampling and/or loss computation. Formally, the layer solves the following two optimization problems for each state $\boldsymbol{s}$

$$\underset{\tilde{\boldsymbol{\mu}}_s}{\arg\min}\, d_{\text{mean}}\left(\tilde{\boldsymbol{\mu}}_s, \boldsymbol{\mu}(s)\right), \quad \text{s.t.} \quad d_{\text{mean}}\left(\tilde{\boldsymbol{\mu}}_s, \boldsymbol{\mu}_{\text{old}}(s)\right) \leq \epsilon_{\boldsymbol{\mu}}, \quad \text{and}$$

$$\underset{\tilde{\boldsymbol{\Sigma}}_s}{\arg\min}\, d_{\text{cov}}\left(\tilde{\boldsymbol{\Sigma}}_s, \boldsymbol{\Sigma}(s)\right), \quad \text{s.t.} \quad d_{\text{cov}}\left(\tilde{\boldsymbol{\Sigma}}_s, \boldsymbol{\Sigma}_{\text{old}}(s)\right) \leq \epsilon_{\Sigma},$$

where $\tilde{\boldsymbol{\mu}}_s$ and $\tilde{\boldsymbol{\Sigma}}_s$ are the optimization variables for input state $\boldsymbol{s}$ and $\epsilon_{\mu}$ and $\epsilon_{\Sigma}$ are the trust region bounds for mean and covariance, respectively. Finally, $d_{\text{mean}}$ and $d_{\text{cov}}$ are the similarity metrics for the mean and covariance of a decomposable distance or divergence measure. Otto et al. [9] proposed three such measures, we will, however, only leverage the decomposed KL-divergence in this work. The trust region for the KL-divergence measure can be made fully differentiable as shown in [9] and is also explained in more detail in Appendix A.

**Reinforcement Learning with Movement Primitives.** While most work on RL with movement primitives [5, 21, 22, 23] concentrates on learning a single MP parameter vector for a single task configuration, a few methods allow linear adaptation [6, 24, 25] of the MP's parameter vector to the context. In terms of step-based deep RL, Bahl et al. [26] propose *neural dynamic processes* (NDP). Their goal is to embed the structure of *dynamical movement primitives* (DMPs) into deep policies by reparametrizing action spaces via second-order differential equations. This can be seen as intersection between step-based and trajectory methods by learning subtrajectories via DMPs spanning multiple timesteps. Their approach allows for effective replanning, but unlike our approach, their main exploration still takes place at the action level rather than at the trajectory level. Hence, equivalent to standard step-based approaches, they have difficulties with sparse and non-Markovian rewards. Lastly, using DMPs requires several numerical integration steps that also need to be differentiated, which is computationally expensive.

# 3 Deep Black-Box Reinforcement Learning

Formally, we learn a policy $\pi_{\boldsymbol{\theta}}(\boldsymbol{w}|\boldsymbol{c})$ that models a distribution over the MP parameters $\boldsymbol{w}$ based on the context information $\boldsymbol{c}$. For most traditional RL tasks, the context space is often a subset of the observations space. As we learn a policy in the parameter space $\boldsymbol{w}$, exploration is also taking place at this level. Instead of sampling actions in each time step, we sample a parameter vector $\boldsymbol{w}$ only in the beginning of each episode. This leads to exploration at the level of desired trajectories and thus to much smoother and temporally correlated exploration. An overview is shown in Figure 1.

## 3.1 Episode-Based Reinforcement Learning Objective

Similar to step-based policy gradient [27, 18], we can also optimize the advantage function using the likelihood ratio gradient and an importance sampling estimator in the ERL setting. Our objective has thus the following form

$$\hat{J}(\pi_{\boldsymbol{\theta}}, \pi_{\boldsymbol{\theta}_{\text{old}}}) = \mathbb{E}_{(\boldsymbol{c}, \boldsymbol{w}) \sim p(\boldsymbol{c}), \pi_{\boldsymbol{\theta}_{\text{old}}}} \left[ \frac{\pi_{\boldsymbol{\theta}}(\boldsymbol{w}|\boldsymbol{c})}{\pi_{\boldsymbol{\theta}_{\text{old}}}(\boldsymbol{w}|\boldsymbol{c})} A^{\pi_{\boldsymbol{\theta}_{\text{old}}}}(\boldsymbol{c}, \boldsymbol{w}) \right],$$

which is maximized w.r.t $\boldsymbol{\theta}$. The term $A^{\pi}(\boldsymbol{c}, \boldsymbol{w}) = \mathbb{E}[R|\boldsymbol{c}, \boldsymbol{w}; \pi] - \mathbb{E}[R|\boldsymbol{c}; \pi]$ denotes the advantage function and $\pi_{\boldsymbol{\theta}_{\text{old}}}$ is the old policy used for sampling. We can further make use of a learned context-value function $V_{\phi}(\boldsymbol{c}) \approx \mathbb{E}[R|\boldsymbol{c}; \pi]$ for the advantage estimator, which is approximated by optimizing

$$\arg\min_{\phi} \mathbb{E}_{(\boldsymbol{c}, \boldsymbol{w}) \sim p(\boldsymbol{c}), \pi_{\boldsymbol{\theta}_{\text{old}}}} \left[ (V_{\phi}(\boldsymbol{c}) - R(\boldsymbol{w}, \boldsymbol{c}))^2 \right].$$

Unlike step-based methods, our approach leverages samples of the form $(\boldsymbol{c}, \boldsymbol{w}, R)$ that are generated per trajectory, where $R$ is the trajectory performance.

## 3.2 Choice of the Deep Reinforcement Learning Algorithm

Training this type of policy can theoretically be done with most existing deep RL methods. Yet, learning in the parameter space requires to learn policies with a higher level of precision than in the step-based case. In the step-based case, smaller errors during action selection can still be corrected at a later time step. In ERL, we only select one parameter vector $\boldsymbol{w}$ per episode, hence no error correction of these parameters is possible. As a consequence, we chose TRPL since this method has shown to be more stable and precise than other RL methods as the trust regions for the policy update are implemented exactly. Furthermore, the trust regions are enforced per state while most other deep RL methods [18, 2, 28] only offer approximate trust region updates for which the trust region is enforced for the mean policy change over all states. We will now present the objective based on the adaptation of the TRPL algorithm to the ERL setup.

## 3.3 Movement Primitives as Parametrized Controllers

So far we only discussed how to learn the distribution in the parameter space, but not how to create actions. For this purpose, we use a trajectory generator $\Psi(\boldsymbol{w})$ that is typically translating the parameters $\boldsymbol{w}$ to a desired trajectory $\boldsymbol{\tau}^d = (\boldsymbol{s}_1^d, \boldsymbol{s}_2^d, \ldots, \boldsymbol{s}_T^d)$. Common choices of trajectory generators are DMPs [29, 30], *probabilistic movement primitives* (ProMPs) [31], or Viapoint MPs [32]. The step-based control signal is then retrieved by employing a trajectory tracking controller $f(\boldsymbol{s}_t, \boldsymbol{s}_t^d)$, such as a simple PD-controller that outputs torques given the current and desired position as well as velocity. Here, *no noise* is applied to the raw actions for exploration. Even more complex controllers such as Cartesian impedance controllers can be readily used.

As a result, trajectory generation and execution is independent of the policy training, enabling the use of any trajectory generator and tracking controller. For this work, we will use ProMPs [31] for trajectory generation and torque-based PD-controllers for action execution. In the case of a ProMP, we can optimize its weight vector that linearly influences the generated trajectories. Further, we can include additional parameters, such as the trajectory starting time as well as the execution speed of the motion (represented as velocity of the ProMP's phase variable) to our ERL optimization process. As our experiments show, this offers a powerful parametrization for tasks where exact timing is crucial, such as robot table tennis and throwing.

### 3.4 Advantages of Episode-Based Versus Step-Based Reinforcement Learning

Step-based RL is currently by far the more prominent perspective on RL. This seems a natural choice as step-based RL exploits the temporal structure of RL problems and is therefore expected to outperform ERL in terms of sample efficiency. Yet, step-based RL also comes with a few disadvantages. First, exploration in action space often results in a very jerky random walk behavior that does not fully explore the trajectory space of the agent. In addition, the stochastic action selection also results in noisy returns, which translates into high variance of the policy gradient estimate. Second, the step-based exploration process also complicates the use of sparse or non-Markovian rewards in the step-based setting. While there do exist specialized solutions for some of these problems, they often require much more complex algorithms.

In contrast, ERL offers a simple framework for learning complex behavior with sparse and non-Markovian rewards, without special treatment of these cases. ERL only uses a fraction of data points to update the policy, since each trajectory is abstracted as only a single data point. While the small number of available training samples may initially appear to be a disadvantage, it offers three distinct benefits: (i) As the used controllers are deterministic, the returns used for the policy updates contain less variance than in the step-based case, which simplifies finding good policy updates. (ii) The exploration in parameter space allows for learning with sparse and non-Markovian rewards which can be used to define the desired behavior in a more direct way. Finally, (iii) the controller parametrization allows the inclusion of temporal parameters such as shifting or scaling of the desired trajectories. We believe that all these features constitute to superior performance of learned ERL policies in comparison to the step-based variant in most of our experiments. In conclusion, we consider ERL as a promising alternative to step-based RL in particularly for tasks where the desired behavior is hard to define with dense rewards.

## 4 Experimental Results

For our evaluation, we first demonstrate the benefits of deep ERL in terms of sparse rewards, precision, and energy efficiency. Afterwards, we conduct a large scale study on all 50 Meta-World tasks [33] to show that we achieve a competitive performance on a variety of robot manipulation tasks with highly shaped dense rewards. Lastly, we investigate multiple challenging control problems that are hard to solve in the step-based setting. We compare our method, which we call *BBRL-TRPL*, to the step-based methods PPO [2], TRPL [9], SAC [1], and NDP [26] as well as to a deep *evolution strategies* (ES) approach [16], the linear adaption method CMORE with ProMPs [8] and BBRL-PPO, which is equivalent to BBRL-TRPL but trained with PPO instead of TRPL.

Note that for NDP, the authors report the performance in terms of the used samples and not in terms of environment interactions (the original work only uses every fifth interaction). We report the total number of environment interactions because we think this leads to a fairer comparison and also explains the rather poor performance of NDP in our experiments. For all tasks the context information $c$ is represented as a subset of the full observation of the first time step including only the stochastic elements, i. e. the parts that are randomly initialized, such as goal or object positions. The trajectory performance $R(w, c)$ is, if not specified otherwise, the full undiscounted trajectory return. We always evaluate on 20 different seeds and compute 10 evaluation runs after each iteration. Following Agarwal et al. [34], we report the *interquartile mean* (IQM) with a 95% stratified bootstrap confidence interval and performance profiles where feasible (for hyperparameters see Appendix D).

### 4.1 Sparse Rewards Induce Energy Efficient Behavior

As introductory task we use an extension of the reacher from OpenAI gym [35]. Instead of two actuated joints, we use five, but limit the context space, i. e. the location of the goal, to $y \geq 0$. This results in an increased control complexity with a slightly decreased task complexity. We investigate two types of rewards, a dense reward equivalent to the original reacher and a sparse reward that only provides the distance to the goal in the last episode time step (see Appendix B.1). While BBRL-TRPL and BBRL-PPO can solve the task for both rewards, NDP and ES do not succeed at either (Figure 2 left and center left). PPO and TRPL achieve a slightly better asymptotic performance than BBRL-TRPL in the dense setting, but are not able to consistently reach the goal for the sparse reward signal. SAC achieves a comparable performance to BBRL-TRPL in the dense setting, however,

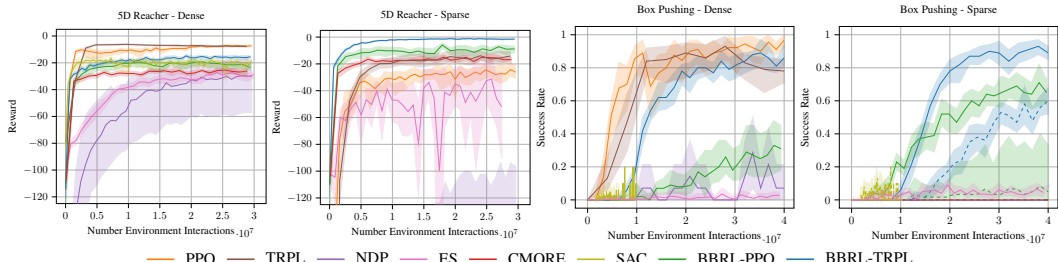

Figure 2: The learning curve for the 5D reacher with dense (left) and sparse reward (center left) signal. The box pushing task is trained with three different rewards. The dense reward (center right) has the best performance for step-based PPO, while it performs poorly with sparsity in time (right, solid line) and sparsity in time and space (right, dashed line). BBRL-TRPL solves the task in the first two settings and is even partially able to do so for the more difficult sparse reward.

cannot leverage the sparse reward (see Appendix C) While, CMORE performs reasonably well, it is only able to cover part of context space the due to its linear adaption.

To demonstrate these learning capabilities for sparse rewards in a more complex setting, we also evaluate the algorithms on a box pushing task. The goal is to move a box to a given goal location and orientation using a simulated Franka Emika Panda (see Appendix B.2). We consider three rewards, a dense reward based on the goal and rotation distance for each time step and two sparse rewards. The first induces time-dependent sparsity by returning the reward signal only in the final time step. The second extends this to include space-dependent sparsity, and additionally requires the box to move near the goal in order to receive the reward. PPO and TRPL can only solve this task in the dense setting (Figure 2 center right and right). SAC and ES even struggle with the dense reward. For SAC, penalties due to constraint violations in particular lead to this unstable behavior. The same applies for NDP, thus we did not evaluate it for the sparse rewards. BBRL-TRPL, on the other hand, learns the task in the dense and the first sparse setting without any problems. Even for the complex time and space sparsity, BBRL-TRPL is able to partially solve the task. BBRL-PPO receives merely moderate results for the dense and first sparse setting.

Yet, why would we use sparse rewards when dense rewards work well with step-based RL? A dense reward typically induces a rather fast motion with poor energy efficiency. As shown in our experiments, dense rewards do not work well if we want to generate accurate motions that reach the goal after a certain time and are energy efficient as well. Here, we can use sparse rewards instead, i. e. , the goal distance is penalized only in the final time step, while the energy cost is accounted for in each time step.

To illustrate the trade-off between precision and energy efficiency, we analyzed the final behaviors in both reward setups with different action penalty factors in the reward function. For each of these factors, we computed the average precision and energy consumption (Figure 3 left and center left). For all methods, decreasing the action penalty factor leads to a higher task precision. However, for the dense reward, a high task precision can only be achieved with high energy costs, while the sparse reward generates behavior of similar precision with one (box pushing) or even two (reacher) factors of magnitude less energy consumption. When analyzing the behaviors, it is visible that the dense reward behavior quickly moves to the target and stays there while the sparse reward behavior reaches the target only slightly before the specified end of the episode, resulting in a much slower, smoother, and more energy efficient motion.

## 4.2 Large Scale Robot Manipulation

On the Meta-World benchmark suite [33], we showcase our ability to learn high quality policies. We train individual policies for each environment, but use the same hyperparameters. PPO and TRPL achieve the best sample complexity (Figure 3 center right). Nevertheless, BBRL-TRPL has competitive asymptotic performance and even converges slightly higher than PPO. While in the aggregated view the gap is rather small, the corresponding performance profiles (Figure 3 right) show that BBRL-TRPL performs better above the 80% threshold. This means that BBRL-TRPL finds

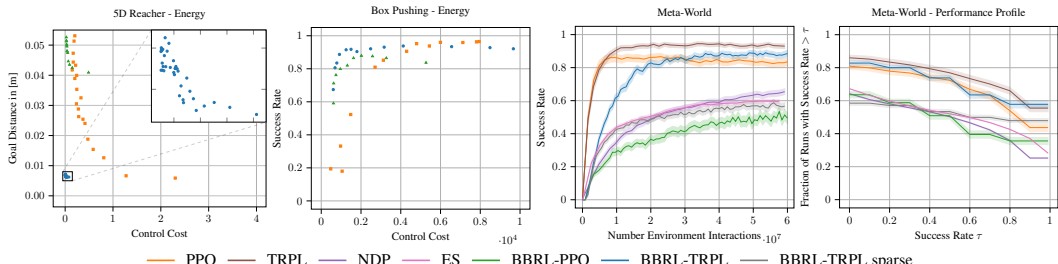

Figure 3: Energy efficiency (sum of squared control action costs) and task performance (distance to the target in the last time step or success rate) trade-off for reacher (left) and box pushing (center left). We average over 100 evaluation runs and all seeds and choose action penalty factors in the intervals $(0, 100]$ for the reacher and $(0, 1]$ for box pushing. For both tasks, BBRL-TRPL with sparse rewards can achieve a much higher energy efficiency (1 to 2 factors of magnitude) while achieving a similar target precision as PPO in the dense reward setting. The success rate (center right) for all 50 Meta-World tasks and the corresponding performance profile (right), i.e. the fraction of runs that perform above the threshold given by the x axis. While the sample efficiency is lower for BBRL-TRPL, the final quality of the policy is higher than PPO.

more consistent solutions than PPO with higher precision and solves these tasks without failures. NDP, ES, and BBRL-PPO are not achieving a competitive performance.

As an additional ablation study, we trained BBRL-TRPL using a sparse reward (only the final step reward of each episode is used), denoted as BBRL-TRPL sparse. While the IQM score is lower, it still completes 50% of the tasks with 100% success rate, which is higher than PPO using the dense reward. Moreover, the slope of the performance profile is rather small, i.e. almost all tasks that can be solved are solved to perfection. After further investigation, we found that poorly performing tasks all require behavior that involves sub-goals, e.g. bin picking, pick and place, etc. This could be addressed e.g. by sequencing several MPs and also giving the corresponding sub-goal as intermediate rewards, which is an interesting direction for future research.

### 4.3 Dealing with non-Markovian Rewards

As final evaluation, we demonstrate the effectiveness of our method in the presence of non-Markovian rewards. These rewards are particularly useful for complex robot learning tasks, where the whole trajectory history is needed to provide feedback to the agent. We first use a modification of the hopper from Open AI gym [35], where we aim to jump as high as possible and land at a target location (see Appendix B.3). The non-Markovian reward returns the maximum height and the minimum distance to the target during the episode. We compare our results to CMORE and ES, which also use the non-Markovian reward. PPO, SAC, and TRPL are trained with a Markovian version that provides height and goal distance in each time step. Please note that we conducted extensive reward shaping to gain the highest performance (i.e. maximum height and minimum goal distance). Overall, BBRL-TRPL achieves a higher jump height than most other methods (Figure 4 left) accompanied with a smaller target distance (Figure 4 center left). BBRL-PPO and CMORE can match the target distance, SAC can even exceed it, but none can reliably learn a good jump height. While BBRL-TRPL charges energy and then jumps just once, the step-based methods try to maximize the height in each time step leading to multiple jumps in one episode (see Appendix C).

Beer pong [25] is another example where non-Markovian rewards are beneficial. The goal is to throw a ball into a cup at various locations on a table. The return is computed from the entire trajectory of the ball, e.g. by leveraging table contacts or the minimum distance to the cup (see Appendix B.4). Since we cannot directly train PPO on such a reward and designing a Markovian version is particularly difficult here, we fix the ball release time and consider the time between release of the ball and the end of the ball trajectory as final time step. This allows to compute the reward in a similar manner as for the non-Markovian setting. We evaluate BBRL-TRPL and CMORE on the non-Markovian reward with a learned ball release time as additional controller parameter as well as PPO for the setting described above (Figure 3 right). BBRL-TRPL and BBRL-PPO both manage to throw the ball into the cup, while PPO struggles. CMORE can reliably throw

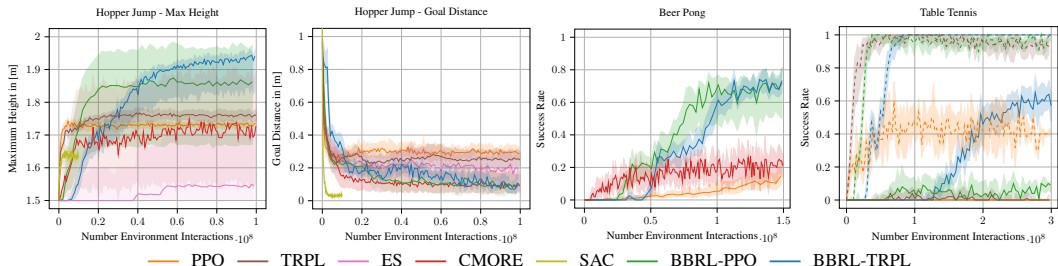

Figure 4: The maximum jumping height of the hopper's center of mass (left) and the target distance (center left). The non-Markovian reward allows to jump approximately 20cm higher with increased goal precision. The beer pong task (center right) demonstrates a similar behavior. PPO has difficulties throwing the ball into the cup even with the fixed optimal release point, whereas BBRL-TRPL can consistently succeed in the tasks with dynamic release points. The success rate of the table tennis task (right) for hitting the ball (dashed line) and successfully returning the ball near the target position (solid line) shows BBRL-TRPL consistently hits the ball and returns it in most of cases.

the ball, however only for a subset of the context space. Similar to the jumping task, BBRL-PPO exhibits a larger confidence interval as it is not able to consistently solve the task.

Lastly, we train an agent for simulated table tennis [25] where the context is four dimensional and given by features of the initial ball trajectory and the desired location for returning the ball (see Appendix B.5). Intuitively, the agent should be rewarded if it hits the ball and returns the ball near the designated goal position. Similar to the throwing task, for the step-based methods we consider the time after hitting the ball as one time step. For the BBRL approaches, we also learn the trajectory starting time as well as the speed of the desired trajectory (which is a parameter of the ProMP). Both parameters help to learn the precise timing required to play table tennis. BBRL-TRPL always manages to hit the ball and successfully returns it within the vicinity of the goal in more than 60% of the cases. While BBRL-PPO and TRPL can hit the ball, they cannot return it with enough precision. We also tested BBRL-TRPL with fixed timing parameters (start time and velocity) and observed a similar behavior - the algorithm requires these timing parameters to learn how to put the ball at the desired location. PPO fails to even hit the ball consistently.

Summarizing, BBRL-TRPL is able to successfully deal with non-Markovian reward structures, which can be more intuitive, easier to define, and achieve better learned behaviors than engineered dense rewards.

## 5 Conclusion and Limitations

In this work, we proposed a new algorithm for ERL. Our method leverages the recently introduced TRPLs [9] to guarantee precise and robust policy updates in high-dimensional parameter spaces. In thorough empirical evaluations, we have shown that we can indeed learn policies with high precision even in the presence of sparse and non-Markovian rewards, where step-based approaches typically fail. Due to the use of sparse rewards, our learned behaviors are more energy efficient and less sensitive to high action costs, which leads to almost universally well-performing policies.

The main limitation of our method and ERL in general is that they typically require more interaction time than step-based RL in the dense reward setting. While we agree this can be critical in some scenarios, we still think the benefits outweigh this drawback in many scenarios. To address the sample complexity issue, we plan to investigate off-policy approaches of our method as well as re-planning of the motion primitive to obtain more training data for the policy update in future work. Another limitation of our method is that the desired trajectory is planned in advance in the beginning of the episode, and hence, cannot be altered during the execution. This might be problematic for unforeseen events or perturbations, i.e. highly complex or reactive behavior that cannot directly be modeled with the current motion representation. We believe that this limitation can also be tackled by incorporating re-planning the MP trajectory in the RL algorithm. For future work, we will also investigate sequencing multiple MPs to solve more complex tasks which are composed of sub-goals.

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

# A   KL-Divergence Trust Region Projection Layer

As already mentioned in the main text, TRPLs [9] present a scalable and mathematically sound approach for enforcing trust regions in step-based deep RL. The layer takes the output of a standard Gaussian policy as input in terms of mean $\boldsymbol{\mu}$ and variance $\boldsymbol{\Sigma}$ and projects it into the trust region if the given mean and variance violate their respective bounds. This projection is done for each input state individually. Subsequently, the projected Gaussian policy distribution with parameters $\tilde{\boldsymbol{\mu}}, \tilde{\boldsymbol{\Sigma}}$ is used for any further steps, e. g. for sampling and/or loss computation. Formally, the layer solves the following two optimization problems for each state $\boldsymbol{s}$

$$\arg\min_{\tilde{\boldsymbol{\mu}}_s} d_{\text{mean}}\left(\tilde{\boldsymbol{\mu}}_s, \boldsymbol{\mu}(s)\right), \quad \text{s.t.} \quad d_{\text{mean}}\left(\tilde{\boldsymbol{\mu}}_s, \boldsymbol{\mu}_{\text{old}}(s)\right) \leq \epsilon_{\boldsymbol{\mu}}, \quad \text{and} \tag{1}$$

$$\arg\min_{\tilde{\boldsymbol{\Sigma}}_s} d_{\text{cov}}\left(\tilde{\boldsymbol{\Sigma}}_s, \boldsymbol{\Sigma}(\boldsymbol{s})\right), \quad \text{s.t.} \quad d_{\text{cov}}\left(\tilde{\boldsymbol{\Sigma}}_s, \boldsymbol{\Sigma}_{\text{old}}(\boldsymbol{s})\right) \leq \epsilon_{\Sigma}, \text{`} \tag{2}$$

where $\tilde{\boldsymbol{\mu}}_s$ and $\tilde{\boldsymbol{\Sigma}}_s$ are the optimization variables for input state $\boldsymbol{s}$ and $\epsilon_{\mu}$ and $\epsilon_{\Sigma}$ are the trust region bounds for mean and covariance, respectively. Finally, $\mu_{\text{old}}$ and $\Sigma_{\text{old}}$ are the reference mean and covariance for the trust region and $d_{\text{mean}}$ as well as $d_{\text{cov}}$ are the similarity metrics for the mean and covariance of a decomposable distance or divergence measure. As we only leverage the KL-divergence projection, we will provide only details for this particular projection below. For the other two projections we refer the reader to Otto et al. [9].

Inserting the mean part of the Gaussian KL divergence into Equation 1 yields

$$\arg\min_{\tilde{\mu}} (\mu - \tilde{\mu})^{\text{T}} \Sigma_{\text{old}}^{-1} (\mu - \tilde{\mu}) \quad \text{s.t.} \quad (\mu_{\text{old}} - \tilde{\mu})^{\text{T}} \Sigma_{\text{old}}^{-1} (\mu_{\text{old}} - \tilde{\mu}) \leq \epsilon_{\mu}.$$

After differentiating the dual w.r.t. $\tilde{\mu}$, we can solve for the projected mean

$$\tilde{\mu} = \frac{\mu + \omega \mu_{\text{old}}}{1 + \omega} \quad \text{with} \quad \omega = \sqrt{\frac{(\mu_{\text{old}} - \mu)^{\text{T}} \Sigma_{\text{old}}^{-1} (\mu_{\text{old}} - \mu)}{\epsilon_{\mu}}} - 1,$$

leveraging the optimal Lagrange multiplier $\omega$. Similarly, we can insert the covariance part of the Gaussian KL divergence into Equation 2, which results in

$$\arg\min_{\tilde{\Sigma}} \text{tr}\left(\Sigma^{-1}\tilde{\Sigma}\right) + \log\frac{|\Sigma|}{|\tilde{\Sigma}|}, \quad \text{s.t.} \quad \text{tr}\left(\Sigma_{\text{old}}^{-1}\tilde{\Sigma}\right) - d + \log\frac{|\Sigma_{\text{old}}|}{|\tilde{\Sigma}|} \leq \epsilon_{\Sigma},$$

where $d$ is the number of degrees of freedom (DoF). Once again, differentiating and solving the dual $g(\eta)$ for the projected covariance yields

$$\tilde{\Sigma} = \left(\frac{\eta^* \Sigma_{\text{old}}^{-1} + \Sigma^{-1}}{\eta^* + 1}\right)^{-1} \quad \text{with} \quad \eta^* = \arg\min_{\eta} g(\eta), \text{ s.t. } \eta \geq 0.$$

Here, the the optimal Lagrange multiplier $\eta^*$ cannot be computed in closed form, however, a standard numerical optimizer, such as BFGS, is able to efficiently find it. This can be made differentiable by taking the differentials of the KKT conditions of the dual. For more details, we refer to the original work [9].

# B   Environment Details

## B.1   Reacher5d

For the Reacher task we modify the original OpenAI gym Reacher-v2 by adding three additional joints, resulting in a total of five joints. The task goal is still to minimize the distance between

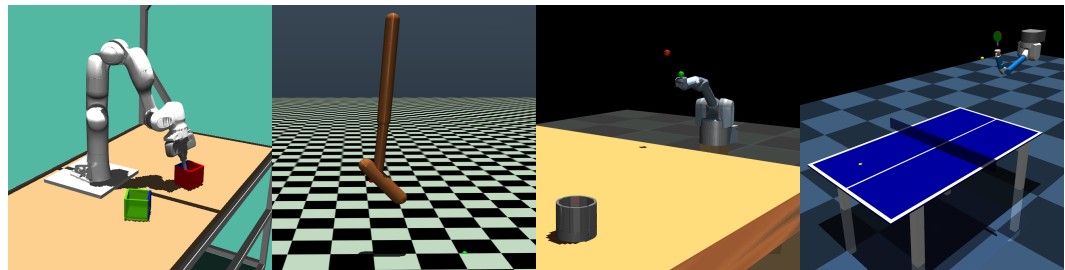

Figure 5: Visualization of the four control tasks box pushing, hopper jumping, beer pong, and table tennis.

the goal point $\mathbf{p}_{goal}$ and the end-effector $\mathbf{p}$. We, however, only sample the goal point for $y \geq 0$, i. e. in the first two quadrants, to slightly reduce task complexity while maintaining the increased control complexity. The observation space remains unchanged, unless for the sparse reward where we additionally add the current step value to make learning possible for step-based methods. The context space only contains the coordinates of the goal position. The action space is the 5d equivalent to the original version.

For the reward the original setting leverages the goal distance

$$R_{\text{goal}} = \|\mathbf{p} - \mathbf{p}_{goal}\|_2$$

and the action cost

$$\tau_t = \sum_i^K (a_t^i)^2,$$

**Dense Reward.** The dense reward in the 5d setting, hence, stays the same and the agent receives in each time step $t$

$$R_{\text{tot}} = -\tau_t - R_{\text{goal}}$$

**Sparse Reward.** The sparse reward only returns the task reward in the last time step $T$ and additionally adds a velocity penalty $R_{\text{vel}} = \sum_i^K (\dot{q}_T^i)^2$, where $\dot{\mathbf{q}}$ are the joint velocities, to avoid overshooting

$$R_{\text{tot}} = \begin{cases} -\tau_t & t < T, \\ -\tau_t - 200 R_{\text{goal}} - 10 R_{\text{vel}} & t = T. \end{cases}$$

### B.2   Box Pushing

The goal of the box pushing task is to move a box to a specified goal location and orientation using the seven DoF Franka Emika Panda. Hence, the context space for this task is the goal position $x \in [0.3, 0.6]$, $y \in [-0.45, 0.45]$ and the goal orientation $\theta \in [0, 2\pi]$. In addition to the contexts, the observation space for the step-based algorithms contains the sine and cosine of the joint angles and their velocities as well as position and orientation quaternions for both end-effector and the box. For the action space we use the torques per joint and additionally add gravity compensation in each time step, that does not have to be learnt. The task is considered successfully solved if the position distance $\leq 0.05$m and the orientation error $\leq 0.5$rad. For the total reward we consider different sub-rewards. First, the distance to the goal

$$R_{\text{goal}} = \|\mathbf{p} - \mathbf{p}_{goal}\|,$$

where $\mathbf{p}$ is the box position and $\mathbf{p}_{goal}$ the goal position itself. Second, the rotation distance

$$R_{\text{rotation}} = \frac{1}{\pi} \arccos |\mathbf{r} \cdot \mathbf{r}_{goal}|,$$

where $\mathbf{r}$ and $\mathbf{r}_{goal}$ are the box orientation and goal orientation in quaternion, respectively. Third, an incentive to keep the rod within the box

$$R_{\text{rod}} = \text{clip}(\|\mathbf{p} - \mathbf{h}_{pos}\|, 0.05, 10)$$

where $\mathbf{h}_{pos}$ is the position of the rod tip. Fourth, a similar incentive that encourages to maintain the rod in a desired rotation

$$R_{\text{rod\_rotation}} = \text{clip}(\frac{2}{\pi}\arccos|\mathbf{h}_{rot}\cdot\mathbf{h}_0|, 0.25, 2),$$

where $\mathbf{h}_{rot}$ and $\mathbf{h}_0 = (0.0, 1.0, 0.0, 0.0)$ are the current and desired rod orientation in quaternion, respectively. And lastly, we utilize the following error

$$\text{err}(\mathbf{q}, \dot{\mathbf{q}}) = \sum_{i\in\{i||q_i|>|q_i^b|\}}(|q_i| - |q_i^b|) + \sum_{j\in\{j||\dot{q}_j|>|\dot{q}_j^b|\}}(|\dot{q}_j| - |\dot{q}_j^b|).$$

Here, $\mathbf{q}$, $\dot{\mathbf{q}}$, $\mathbf{q}^b$, and $\dot{\mathbf{q}}^b$ are the robot joint's position and velocity as well as their respective bounds. Additionally, we consider an action cost in each time step $t$

$$\tau_t = 5\cdot 10^{-4}\sum_i^K (a_t^i)^2,$$

where $K = 7$ is the number of DoF. In total we consider three different rewards.

**Dense Reward.** The dense reward provides information about the goal and rotation distance in each time step $t$ on top of the utility rewards

$$R_{\text{tot}} = -R_{\text{rod}} - R_{\text{rod\_rotation}} - \tau_t - \text{err}(\boldsymbol{q}, \dot{\boldsymbol{q}}) - 3.5R_{\text{goal}} - 2R_{\text{rotation}}.$$

**Time-Dependent Sparse Reward.** The time-dependent sparse reward is similar to the dense reward, but only returns the goal and rotation distance in the last time step $T$

$$R_{\text{tot}} = \begin{cases} -R_{\text{rod}} - R_{\text{rod\_rotation}} - \tau_t - \text{err}(\mathbf{q}, \dot{\mathbf{q}}), & t < T, \\ -R_{\text{rod}} - R_{\text{rod\_rotation}} - \tau_t - \text{err}(\mathbf{q}, \dot{\mathbf{q}}) - 350R_{\text{goal}} - 200R_{\text{rotation}}, & t = T. \end{cases}$$

**Time- and Space-Dependent Sparse Reward.** The second sparse reward additionally adds sparsity based on the position and only returns goal and rotation distance in the last time step when the box is near the goal location

$$R_{\text{tot}} = \begin{cases} -R_{\text{rod}} - R_{\text{rod\_rotation}} - \tau_t - \text{err}(\mathbf{q}, \dot{\mathbf{q}})\cdots \\ \cdots - \text{clip}(1050R_{\text{goal}}, 0, 100) - \text{clip}(15R_{\text{rotation}}, 0, 100) + 300, & t = T \text{ and } R_{\text{goal}} \le 0.1, \\ -R_{\text{rod}} - R_{\text{rod\_rotation}} - \tau_t - \text{err}(\mathbf{q}, \dot{\mathbf{q}}), & \text{else.} \end{cases}$$

### B.3 Hopper Jump

In the Hopper jump task the agent has to learn to jump as high as possible and land on a certain goal position at the same time. We consider five basis functions per joint resulting in an 15 dimensional weight space. The context is four-dimensional consisting of the initial joint angles $\theta \in [-0.5, 0]$, $\gamma \in [-0.2, 0]$, $\phi \in [0, 0.785]$ and the goal landing position $x \in [0.3, 1.35]$. The full observation space extends the original observation space from the OpenAI gym Hopper by adding the x-value of the goal position and the x-y-z difference between the goal point and the reference point at the Hopper's foot. The action space is the same as for the original Hopper task. We consider a non-Markovian reward function for the episode-based algorithms and a step-based reward for PPO, which we have extensively designed to obtain the highest possible jump.

**Non-Markovian Reward.** In each time-step $t$ we provide an action cost

$$\tau_t = 10^{-3}\sum_i^K (a_t^i)^2,$$

where $K = 3$ is the number of DoF. In the last time-step $T$ of the episode we provide a reward which contains information about the whole episode as

$$\begin{aligned} R_{height} &= 10h_{max}, \\ R_{gdist} &= ||p_{foot,T} - p_{goal}||_2, \\ R_{cdist} &= ||p_{foot,contact} - p_{goal}||_2, \\ R_{healthy} &= \begin{cases} 2 & \text{if } z_T \in [0.5, \infty]\text{and }\theta, \gamma, \phi \in [-\infty, \infty] \\ 0 & \text{else} \end{cases}, \end{aligned}$$

where $h_{max}$ is the maximum jump height in z-direction of the center of mass reached during the whole episode, $p_{foot,t}$ is the x-y-z position of the foot's heel at time step $t$, $p_{foot,contact}$ is the foot's heel position when having a contact with the ground after the first jump, $p_{goal}$ is the goal landing position of the heel. $R_{healthy}$ is a slightly modified reward of the healthy reward defined in the original hopper task. The hopper is considered as 'healthy' if the z position of the center of mass is within the range $[0.5m, \infty]$. This encourages the hopper to stand at the end of the episode. Note that all states need to be within the range $[-100, 100]$ for $R_{healthy}$. Since this is defined in the hopper task from OpenAI already, we haven't mentioned it here. The total reward at the end of an episode is given as

$$R_{tot} = -\sum_{t=0}^{T} \tau_t + R_{height} + R_{gdist} + R_{cdist} + R_{healthy}.$$

**Step-Based Reward.** We consider a step-based alternative reward such that PPO is also able to learn a meaningful behavior on this task. We have tuned the reward such that we can obtain the best performance. The observation space is the same as in the original hopper task from OpenAI extended with the goal landing position and the current distance of the foot's heel and the goal landing postion. We again consider the action cost in each time-step $t$

$$\tau_t = 10^{-3} \sum_{i}^{K} (a_t^i)^2,$$

and additionally consider the rewards

$$R_{height,t} = 3h_t$$
$$R_{gdist,t} = 3||p_{foot,t} - p_{goal}||_2$$
$$R_{healthy,t} = \begin{cases} 1 & \text{if } z_t \in [0.5, \infty] \text{and } \theta, \gamma, \phi \in [-\infty, \infty] \\ 0 & \text{else} \end{cases},$$

where these rewards are now returned to the agent in each time-step $t$, resulting in the reward per time-step

$$r_t(s_t, a_t) = -\tau_t + R_{height,t} + R_{gdist,t} + R_{healthy,t}.$$

## B.4 Beer Pong

In the Beer Pong task the $K = 7$ Degrees of Freedom (DoF) robot has to throw a ball into a cup on a big table. The context is defined by the cup's two dimensional position on the table which lies in the range $x \in [-1.42, 1.42]$, $y \in [-4.05, -1.25]$. For the step-based algorithms we consider cosine and sine of the robot's joint angles, the angle velocities, the ball's distance to the bottom of the cup, the ball's distance to the top of the cup, the cup position and the current time step. The action space for the step-based algorithms is defined as the torques for each joint, the parameter space for the episode-based methods is 15 dimensional which consists of the two weights for the basis functions per joint and the duration of the throwing trajectory, i.e. the ball release time.

We generally consider action penalties

$$\tau_t = \frac{1}{K} \sum_{i}^{K} (a_t^i)^2,$$

consisting of the sum of squared torques per joint. For $t < T$ we consider the reward

$$r_t(s_t, a_t) = -\alpha_t \tau_t,$$

with $\alpha_t = 10^{-2}$. For $t = T$ we consider the non-Markovian reward

$$R_{task} = \begin{cases} -4 - min(||p_{c,top} - p_{b,1:T}||_2^2) - 0.5||p_{c,bottom} - p_{b,T}||_2^2 \cdots \\ \cdots - 2||p_{c,bottom} - p_{b,k}||_2^2 - \alpha_T \tau, & \text{if cond. 1} \\ -4 - min(||p_{c,top} - p_{b,1:T}||_2^2) - 0.5||p_{c,bottom} - p_{b,T}||_2^2 - \alpha_T \tau, & \text{if cond. 2} \\ -2 - min(||p_{c,top} - p_{b,1:T}||_2^2) - 0.5||p_{c,bottom} - p_{b,T}||_2^2 - \alpha_T \tau, & \text{if cond. 3} \\ -||p_{c,bottom} - p_{b,T}||_2^2 - \alpha_T \tau, & \text{if cond. 4} \end{cases}$$

$$R_{task} = \begin{cases} -4 - min(||p_{c,top} - p_{b,1:T}||_2^2) - 0.5||p_{c,bottom} - p_{b,T}||_2^2 \cdots \\ \quad \cdots - 2||p_{c,bottom} - p_{b,k}||_2^2 - \alpha_T \tau, & \text{if cond. 1} \\ -4 - min(||p_{c,top} - p_{b,1:T}||_2^2) - 0.5||p_{c,bottom} - p_{b,T}||_2^2 - \alpha_T \tau, & \text{if cond. 2} \\ -2 - min(||p_{c,top} - p_{b,1:T}||_2^2) - 0.5||p_{c,bottom} - p_{b,T}||_2^2 - \alpha_T \tau, & \text{if cond. 3} \\ -||p_{c,bottom} - p_{b,T}||_2^2 - \alpha_T \tau, & \text{if cond. 4} \end{cases},$$

where $p_{c,top}$ is the position of the top edge of the cup, $p_{c,bottom}$ is the ground position of the cup, $p_{b,t}$ is the position of the ball at time point $t$, and $\tau$ is the squared mean torque over all joints during one rollout and $\alpha_T = 10^{-4}$. The different conditions are:

- cond. 1: The ball had a contact with the ground before having a contact with the table.
- cond. 2: The ball is not in the cup and had no table contact
- cond. 3: The ball is not in the cup and had table contact
- cond. 4: The ball is in the cup.

Note that $p_{b,k}$ is the ball's and the ground's contact position and is only given, if the ball had a contact with the ground first.

At time step $t = T$ we also give information whether the agent's chosen ball release time $B$ was reasonable

$$R_{release} = \begin{cases} -30 - 10(B - B_{min})^2, & \text{if } B < B_{min} \\ -30 - 10(B - B_{max})^2, & \text{if } B < B_{max} \end{cases},$$

where we define $B_{min} = 0.1s$ and $B_{max} = 1s$, such that the agent is encouraged to throw the ball within the time range $[B_{min}, B_{max}]$.

The total return over the whole episode is therefore given as

$$R_{tot} = \sum_{t=1}^{T-1} r_t(s_t, a_t) + R_{task} + R_{release}$$

A throw is considered as successful if the ball is in the cup at the end of an episode.

## B.5   Table Tennis

We consider table tennis for the entire table, i.e. incoming balls are anywhere on the side of the robot and goal locations anywhere on the opponents side. The goal is to use the seven degree of freedom robotic arm to hit the incoming ball based on its landing position and return it as close as possible to the specified goal location. As context space we consider the initial ball position $x \in [-1, -0.2]$, $y \in [-0.65, 0.65]$ and the goal position $x \in [-1.2, -0.2]$, $y \in [-0.6, 0.6]$. The full observation space again contains cosine and sine of the joint space and the joint velocities as well as the ball velocity on top of the above context information. The torques of the joints make up the action space. For this experiment, we do not use any gravity compensation and allow in the episode-based setting to learn the start time $t_0$ and the trajectory duration $T$. The task is considered successful if the returned ball lands on the opponent's side of table and within $\leq 0.2m$ to the goal location. The reward is defined as

$$r_{task} = \begin{cases} 0, & \text{if cond. 1} \\ 0.2 - 0.2 \tanh(\min ||\mathbf{p}_r - \mathbf{p}_b||^2), & \text{if cond. 2} \\ 3 - 2 \tanh(\min ||\mathbf{p}_r - \mathbf{p}_b||^2) - \tanh(||\mathbf{p}_l - \mathbf{p}_{goal}||^2), & \text{if cond. 3} \\ 6 - 2 \tanh(\min ||\mathbf{p}_r - \mathbf{p}_b||^2) - 4 \tanh(||\mathbf{p}_l - \mathbf{p}_{goal}||^2), & \text{if cond. 4} \\ 7 - 2 \tanh(\min ||\mathbf{p}_r - \mathbf{p}_b||^2) - 4 \tanh(||\mathbf{p}_l - \mathbf{p}_{goal}||^2), & \text{if cond. 5} \end{cases}$$

where $\mathbf{p}_r$ is the position of racket center, $\mathbf{p}_b$ is the position of the ball, $\mathbf{p}_l$ is the ball landing position, $\mathbf{p}_{goal}$ is the target position. The different conditions are

- cond. 1: the end of episode is not reached,
- cond. 2: the end of episode is reached,

- cond. 3: cond.2 is satisfied and robot did hit the ball,
- cond. 4: cond.3 is satisfied and the returned ball landed on the table,
- cond. 5: cond.4 is satisfied and the landing position is at the opponent's side.

The episode ends when any of the following conditions are met

- the maximum horizon length is reached
- ball did land on the floor without hitting
- ball did land on the floor or table after hitting

For BBRL-PPO and BBRL-TRPL, the whole desired trajectory is obtained ahead of environment interaction, making use of this property we can collect some samples without physical simulation. The reward function based on this desired trajectory is defined as

$$r_{traj} = -\sum_{(i,j)} |\tau_{ij}^d| - |q_j^b|, \quad (i,j) \in \{(i,j) \mid |\tau_{ij}^d| > |q_j^b|\}$$

where $\tau^d$ is the desired trajectory, $i$ is the time index, $j$ is the joint index, $q^b$ is the joint position upperbound. The desired trajectory is considered as invalid if $r_{traj} < 0$, an invalid trajectory will not be executed by robot. The overall reward for BBRL is defined as:

$$r = \begin{cases} r_{traj}, & r_{traj} < 0 \\ r_{task}, & \text{otherwise} \end{cases}$$

## C   Additional Evaluations

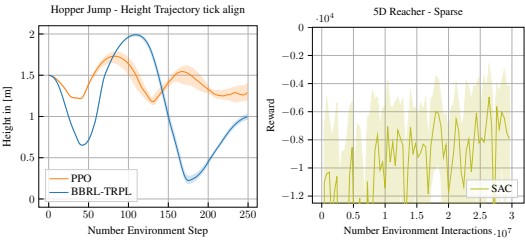

Figure 6: (Left) The improved performance on the Hopper Jump task is also demonstrated on the jumping profile for a fixed context. While BBRL-TRPL jumps once as high as possible, PPO constantly tries to maximize the height at each time step which leads to several jumps throughout the episode and consequently to a lower maximum height. (Right) Learning curve of SAC for the sparse reward of the 5D Reacher task.

# D  Hyperparameters

For all methods, where applicable, we optimized the learning rate, sample size, batch size, number of layers, and the number of epochs. For all BBRL methods and NDP, we additionally optimized the number of basis functions. Moreover, we found that NDP requires tuning of the scale of the predicted DMP weights, which was hard-coded to 100 in the original code base. However, this value only worked for the meta-world tasks, but not for the other tasks, hence we adjusted it to allow for a fair comparison. The population size of ES is always half the number of samples because two function evaluations are used per parameter vector.

Table 1: Hyperparameters for the modified reacher experiments.

| | PPO | NDP | TRPL | SAC | CMORE | ES | BBRL-PPO | BBRL-TRPL |
|---|---|---|---|---|---|---|---|---|
| number samples | 16000 | 16000 | 16000 | 1000 | 120 | 200 | 64 | 64 |
| GAE $\lambda$ | 0.95 | 0.95 | 0.95 | 0.95 | n.a. | n.a. | n.a. | n.a. |
| discount factor | 0.99 | 0.99 | 0.99 | 0.99 | n.a. | n.a. | n.a. | n.a. |
| $\epsilon_\mu/\epsilon$ | n.a. | n.a. | 0.005 | n.a. | 0.1 | n.a. | n.a. | 0.05 |
| $\epsilon_\Sigma$ | n.a. | n.a. | 0.0005 | n.a. | n.a. | n.a. | n.a. | 0.0005 |
| optimizer | adam | adam | adam | adam | n.a. | adam | adam | adam |
| epochs | 10 | 10 | 20 | 1000 | n.a. | n.a. | 100 | 100 |
| learning rate | 3e-4 | 3e-4 | 5e-5 | 3e-4 | n.a. | 1e-2 | 3e-4 | 3e-4 |
| use critic | True | True | True | True | False | False | False | False |
| epochs critic | 10 | 10 | 10 | 1000 | n.a. | n.a. | n.a. | n.a. |
| learning rate critic (and alpha) | 3e-4 | 3e-4 | 3e-4 | 3e-4 | n.a. | n.a. | n.a. | n.a. |
| number minibatches | 32 | 32 | 64 | n.a. | n.a. | n.a. | 1 | 1 |
| batch size | n.a. | n.a. | n.a. | 256 | n.a. | n.a. | n.a. | n.a. |
| buffer size | n.a. | n.a. | n.a. | 1e6 | n.a. | n.a. | n.a. | n.a. |
| learning starts | 0 | 0 | 0 | 10000 | 0 | 0 | 0 | 0 |
| polyak_weight | n.a. | n.a. | n.a. | 5e-3 | n.a. | n.a. | n.a. | n.a. |
| trust region loss weight | n.a. | n.a. | 10 | n.a. | n.a. | n.a. | n.a. | 10 |
| normalized observations | True | True | True | False | False | False | False | False |
| normalized rewards | True | True | False | False | False | False | False | False |
| observation clip | 10.0 | 10.0 | n.a. | n.a. | n.a. | n.a. | n.a. | n.a. |
| reward clip | 10.0 | 10.0 | n.a. | n.a. | n.a. | n.a. | n.a. | n.a. |
| critic clip | 0.2 | 0.2 | n.a. | n.a. | n.a. | n.a. | 0.2 | n.a. |
| importance ratio clip | 0.2 | 0.2 | n.a. | n.a. | n.a. | n.a. | 0.2 | n.a. |
| hidden layers | [32, 32] | [32, 32] | [32, 32] | [128,128] | n.a. | [32, 32] | [32, 32] | [32, 32] |
| hidden layers critic | [32, 32] | [32, 32] | [32, 32] | [128,128] | n.a. | n.a. | n.a. | n.a. |
| hidden activation | tanh | tanh | tanh | relu | n.a. | tanh | tanh | tanh |
| initial std | 1.0 | 1.0 | 1.0 | 1.0 | 1.0 | 1.0 | 1.0 | 1.0 |
| number basis functions | n.a. | 5 | n.a. | n.a. | 5 | n.a. | 5 | 5 |
| number zero basis | n.a. | n.a. | n.a. | n.a. | 1 | n.a. | 1 | 1 |
| weight scale | n.a. | 20 | n.a. | n.a. | n.a. | n.a. | n.a. | n.a. |

Table 2: Hyperparameters for the box pushing experiments.

| | PPO | NDP | TRPL | SAC | ES | BBRL-PPO | BBRL-TRPL |
|---|---|---|---|---|---|---|---|
| number samples | 16000 | 16000 | 16000 | 1000 | 250 | 160 | 160 |
| GAE $\lambda$ | 0.95 | 0.95 | 0.95 | 0.95 | n.a. | n.a. | n.a. |
| discount factor | 0.99 | 0.99 | 0.99 | 0.99 | n.a. | n.a. | n.a. |
| $\epsilon_\mu$ | n.a. | n.a. | 0.005 | n.a. | n.a. | n.a. | 0.005 |
| $\epsilon_\Sigma$ | n.a. | n.a. | 0.00005 | n.a. | n.a. | n.a. | 0.0005 |
| optimizer | adam | adam | adam | adam | adam | adam | adam |
| epochs | 10 | 10 | 20 | 1000 | n.a. | 100 | 100 |
| learning rate | 1e-4 | 1e-4 | 5e-5 | 1e-4 | 1e-2 | 1e-4 | 1e-4 |
| use critic | True | True | True | True | False | True | True |
| epochs critic | 10 | 10 | 10 | 1000 | n.a. | 100 | 100 |
| learning rate critic (and alpha) | 1e-4 | 1e-4 | 2e-4 | 1e-4 | n.a. | 1e-4 | 1e-4 |
| number minibatches | 40 | 32 | 40 | n.a. | n.a. | 1 | 1 |
| batch size | n.a. | n.a. | n.a. | 256 | n.a. | n.a. | n.a. |
| buffer size | n.a. | n.a. | n.a. | 1e6 | n.a. | n.a. | n.a. |
| learning starts | 0 | 0 | 0 | 10000 | 0 | 0 | 0 |
| polyak_weight | n.a. | n.a. | n.a. | 5e-3 | n.a. | n.a. | n.a. |
| trust region loss weight | n.a. | n.a. | 10 | n.a. | n.a. | n.a. | 25 |
| normalized observations | True | True | True | False | False | False | False |
| normalized rewards | True | True | False | False | False | False | False |
| observation clip | 10.0 | 10.0 | n.a. | n.a. | n.a. | n.a. | n.a. |
| reward clip | 10.0 | 10.0 | n.a. | n.a. | n.a. | n.a. | n.a. |
| critic clip | 0.2 | 0.2 | n.a. | n.a. | n.a. | 0.2 | n.a. |
| importance ratio clip | 0.2 | 0.2 | n.a. | n.a. | n.a. | 0.2 | n.a. |
| hidden layers | [256, 256] | [256, 256] | [256, 256] | [256, 256] | [256, 256] | [128, 128] | [128, 128] |
| hidden layers critic | [256, 256] | [256, 256] | [256, 256] | [256, 256] | n.a. | [32, 32] | [32, 32] |
| hidden activation | tanh | tanh | tanh | relu | tanh | tanh | relu |
| initial std | 1.0 | 1.0 | 1.0 | 1.0 | 1.0 | 1.0 | 1.0 |
| number basis functions | n.a. | 5 | n.a. | n.a. | n.a. | 5 | 5 |
| number zero basis | n.a. | n.a. | n.a. | n.a. | n.a. | 1 | 1 |
| weight scale | n.a. | 10 | n.a. | n.a. | n.a. | n.a. | n.a. |

Table 3: Hyperparameters for the Meta-World experiments.

| | PPO | NDP | TRPL | ES | BBRL-PPO | BBRL-TRPL |
|---|---|---|---|---|---|---|
| number samples | 16000 | 16000 | 16000 | 200 | 16 | 16 |
| GAE $\lambda$ | 0.95 | 0.95 | 0.95 | n.a. | n.a. | n.a. |
| discount factor | 0.99 | 0.99 | 0.99 | n.a. | n.a. | n.a. |
| $\epsilon_\mu$ | n.a. | n.a. | 0.005 | n.a. | n.a. | 0.005 |
| $\epsilon_\Sigma$ | n.a. | n.a. | 0.0005 | n.a. | n.a. | 0.0005 |
| optimizer | adam | adam | adam | adam | adam | adam |
| epochs | 10 | 10 | 20 | n.a. | 100 | 100 |
| learning rate | 3e-4 | 3e-4 | 5e-5 | 1e-2 | 3e-4 | 3e-4 |
| use critic | True | True | True | False | False | False |
| epochs critic | 10 | 10 | 10 | n.a. | n.a. | n.a. |
| learning rate critic (and alpha) | 3e-4 | 3e-4 | 3e-4 | n.a. | n.a. | n.a. |
| number minibatches | 32 | 32 | 64 | n.a. | 1 | 1 |
| trust region loss weight | n.a. | n.a. | 10.0 | n.a. | n.a. | 10 |
| normalized observations | True | True | True | False | False | False |
| normalized rewards | True | True | False | False | False | False |
| observation clip | 10.0 | 10.0 | n.a. | n.a. | n.a. | n.a. |
| reward clip | 10.0 | 10.0 | n.a. | n.a. | n.a. | n.a. |
| critic clip | 0.2 | 0.2 | n.a. | n.a. | 0.2 | n.a. |
| importance ratio clip | 0.2 | 0.2 | n.a. | n.a. | 0.2 | n.a. |
| hidden layers | [128, 128] | [128, 128] | [128, 128] | [128, 128] | [32, 32] | [32, 32] |
| hidden layers critic | [128, 128] | [128, 128] | [128, 128] | n.a. | n.a. | n.a. |
| hidden activation | tanh | tanh | tanh | tanh | tanh | relu |
| initial std | 1.0 | 1.0 | 1.0 | 1.0 | 1.0 | 10.0 |
| number basis functions | n.a. | 5 | n.a. | n.a. | 5 | 5 |
| number zero basis | n.a. | n.a. | n.a. | n.a. | 1 | 1 |
| weight scale | n.a. | 100 | n.a. | n.a. | n.a. | n.a. |

Table 4: Hyperparameters for the hopper jumping experiments.

| | PPO | TRPL | SAC | CMORE | ES | BBRL-PPO | BBRL-TRPL |
|---|---|---|---|---|---|---|---|
| number samples | 16384 | 16384 | 1000 | 60 | 200 | 64 | 64 |
| GAE $\lambda$ | 0.95 | 0.95 | 0.95 | n.a. | n.a. | n.a. | n.a. |
| discount factor | 0.99 | 0.99 | 0.99 | n.a. | n.a. | n.a. | n.a. |
| $\epsilon_\mu/\epsilon$ | n.a. | 0.005 | n.a. | 0.1 | n.a. | n.a. | 0.005 |
| $\epsilon_\Sigma$ | n.a. | 0.00005 | n.a. | n.a. | n.a. | n.a. | 0.0005 |
| optimizer | adam | adam | adam | n.a. | adam | adam | adam |
| epochs | 10 | 20 | 1000 | n.a. | n.a. | 100 | 100 |
| learning rate | 3e-4 | 5e-5 | 1e-4 | n.a. | 0.01 | 1e-4 | 5e-5 |
| use critic | True | True | True | False | False | False | False |
| epochs critic | 10 | 10 | 1000 | n.a. | n.a. | n.a. | n.a. |
| learning rate critic (and alpha) | 3e-4 | 3e-4 | 1e-4 | n.a. | n.a. | n.a. | n.a. |
| number minibatches | 32 | 64 | n.a. | n.a. | n.a. | 1 | 1 |
| batch size | n.a. | n.a. | 256 | n.a. | n.a. | n.a. | n.a. |
| buffer size | n.a. | n.a. | 1e6 | n.a. | n.a. | n.a. | n.a. |
| learning starts | 0 | 0 | 10000 | 0 | 0 | 0 | 0 |
| polyak_weight | n.a. | n.a. | 5e-3 | n.a. | n.a. | n.a. | n.a. |
| trust region loss weight | n.a. | 10 | n.a. | n.a. | n.a. | n.a. | 25 |
| normalized observations | True | True | False | False | False | False | False |
| normalized rewards | True | False | False | False | False | False | False |
| observation clip | 10.0 | n.a. | n.a. | n.a. | n.a. | n.a. | n.a. |
| reward clip | 10.0 | n.a. | n.a. | n.a. | n.a. | n.a. | n.a. |
| critic clip | 0.2 | n.a. | n.a. | n.a. | n.a. | 0.2 | n.a. |
| importance ratio clip | 0.2 | n.a. | n.a. | n.a. | n.a. | 0.2 | n.a. |
| hidden layers | [128, 128] | [128, 128] | [128, 128] | n.a | [128, 128] | [32, 32] | [32, 32] |
| hidden layers critic | [128, 128] | [128, 128] | [128, 128] | n.a | n.a | n.a | n.a |
| hidden activation | tanh | tanh | relu | n.a. | tanh | tanh | tanh |
| initial std | 1.0 | 1.0 | 1.0 | 1.0 | 1.0 | 1.0 | 1.0 |
| number basis functions | n.a. | n.a. | n.a. | 5 | n.a. | 5 | 5 |
| number zero basis | n.a. | n.a. | n.a. | 1 | n.a. | 1 | 1 |

Table 5: Hyperparameters for the Beer Pong experiments.

| | PPO | CMORE | BBRL-PPO | BBRL-TRPL |
|---|---|---|---|---|
| number samples | 16384 | 60 | 160 | 160 |
| GAE $\lambda$ | 0.95 | n.a. | n.a. | n.a. |
| discount factor | 0.99 | n.a. | n.a. | n.a. |
| $\epsilon_\mu/\epsilon$ | n.a. | 0.1 | n.a. | 0.005 |
| $\epsilon_\Sigma$ | n.a. | n.a. | n.a. | 0.0005 |
| optimizer | adam | n.a. | adam | adam |
| epochs | 10 | n.a. | 100 | 100 |
| learning rate | 3e-4 | n.a. | 1e-4 | 5e-5 |
| use critic | True | False | False | False |
| epochs critic | 10 | n.a. | n.a. | n.a. |
| learning rate critic (and alpha) | 3e-4 | n.a. | n.a. | n.a. |
| number minibatches | 32 | n.a. | 1 | 1 |
| trust region loss weight | n.a. | n.a. | n.a. | 25 |
| normalized observations | True | False | False | False |
| normalized rewards | True | False | False | False |
| observation clip | 10.0 | n.a. | n.a. | n.a. |
| reward clip | 10.0 | n.a. | n.a. | n.a. |
| critic clip | 0.2 | n.a. | 0.2 | n.a. |
| importance ratio clip | 0.2 | n.a. | 0.2 | n.a. |
| hidden layers | [128, 128] | n.a. | [32, 32] | [32, 32] |
| hidden layers critic | [128, 128] | n.a. | n.a. | n.a. |
| hidden activation | tanh | n.a. | tanh | tanh |
| initial std | 1.0 | 1.0 | 1.0 | 1.0 |
| number basis functions | n.a. | 2 | 2 | 2 |
| number zero basis | n.a. | 2 | 2 | 2 |

Table 6: Hyperparameters for the Table Tennis experiments.

|  | PPO | TRPL | BBRL-PPO | BBRL-TRPL |
|---|---|---|---|---|
| number samples | 16000 | 16000 | 200 | 200 |
| GAE $\lambda$ | 0.95 | 0.95 | n.a. | n.a. |
| discount factor | 0.99 | 0.99 | n.a. | n.a. |
| $\epsilon_\mu$ | n.a. | 0.0005 | n.a. | 0.0005 |
| $\epsilon_\Sigma$ | n.a. | 0.00005 | n.a. | 0.00005 |
| optimizer | adam | adam | adam | adam |
| epochs | 10 | 20 | 100 | 100 |
| learning rate | 1e-4 | 5e-5 | 1e-4 | 3e-4 |
| use critic | True | True | True | True |
| epochs critic | 10 | 10 | 100 | 100 |
| learning rate critic (and alpha) | 1e-4 | 1e-4 | 1e-4 | 3e-4 |
| number minibatches | 40 | 40 | 1 | 1 |
| trust region loss weight | n.a. | 10.0 | n.a. | 25 |
| normalized observations | True | True | False | False |
| normalized rewards | True | False | False | False |
| observation clip | 10.0 | n.a. | n.a. | n.a. |
| reward clip | 10.0 | n.a. | n.a. | n.a. |
| critic clip | 0.2 | n.a. | 0.2 | n.a. |
| importance ratio clip | 0.2 | n.a. | 0.2 | n.a. |
| hidden layers | [256, 256] | [256, 256] | [256] | [256] |
| hidden layers critic | [256, 256] | [256, 256] | n.a. | n.a. |
| hidden activation | tanh | tanh | tanh | relu |
| initial std | 1.0 | 1.0 | 1.0 | 1.0 |
| number basis functions | n.a. | n.a. | 3 | 3 |
| number zero basis | n.a. | n.a. | 1 | 1 |

