# OpenReview forum: "Deep Black-Box Reinforcement Learning with Movement Primitives"
_robot-learning.org/CoRL/2022/Conference — CoRL 2022 Poster_

### Official Review · Reviewer_Rejd · 2022-07-21

**Originality:** Fair
**Technical Quality:** Fair
**Clarity Of Presentation:** Very Good
**Impact:** 3

**Recommendation:**

Weak Reject: I recommend rejecting the paper, but will not argue for my recommendation if the majority of other reviewers have a different opinion.

**Summary:**

The paper presents a novel method for episode-based reinforcement learning (RL), BBRL-TRPL, that uses trust region projection layers (TRPLs) to learn motion parameters, and uses Probabilistic Motion Primitives (ProMP) to generate motor commands. Compared to step-based RL algorithms, BBRL-TRPL shows better performance when the reward is sparse and non-Markovian, and learns more energy-efficient trajectories.

**Issues:**

First, I would like the authors to conduct additional results to benchmark their algorithms against ES baselines (see Weakness sections for more details). In addition, I have a few minor comments on writing:
* Separate Related Works and Background sections.
I found that Section 2 can be too long right now, and it can be confusing to read related works and the priliminary stuff (TRPL, RL) in the same section. It would be nicer to have a dedicated preliminary section for the problem setup.

* Equation in section 3.1: looks like the definition of the cost function is a bit off? The $\arg\max$ should appera in policy optimization, but not in the definition of $J$?

**Quality Of The Limitations Section:**

Limitations are addressed clearly

**Reviewer Expertise:**

3: The reviewer is fairly confident that the evaluation is correct

**Robotics Focus:**

Highly relevant to robotics but no hardware experiments

**Strengths And Weaknesses:**


**Strengths:**

Overall, I find the writing to be clear and easy to follow. I do have a few suggestions on how to improve presentation, but the paper is overall easy to read. Moreover, the authors did thorough evaluation of their method against existing baselines with detailed analysis.

**Weakness:**

My major concern for the paper is that it misses an entire family of papers that use evolutionary strategies (ES) for reinforcement learning tasks, such as CMAES [1] and ARS [2]. The authors mentioned that these algorithms may be limited to "linear adaptations" (Line 43). It would be great if the authors could clarify on what "linear" means here, since these algorithms have been used to find the parameters of neural network policies. Moreover, these algorithms are known to work well across a wide variety of reinforcement learning tasks. Therefore, the authors should at least compare their method some ES baselines. A simple ES baseline could be a single policy that directly takes in the context and the robot states ($c$ and $s_t$ in Fig.1) and outputs motor commands.

Moreover, the major benefits of the proposed algorithm, such as better performance at sparse or non-Markovian rewards and structured exploration, have been known as the benefits of black-box algorithms _in general_ [3]. All these algorithms adapt the policy based on the total _return_ of the entire episode, and are not particularly sensitive to the shape of reward functions. Moreover, they all explore in the parameter space, and outputs smooth actions during training. Therefore, I find that the results of the proposed method does not improve much from existing methods.

[1] The CMA Evolution Strategy: A Tutorial [link](https://arxiv.org/abs/1604.00772)

[2] Simple random search provides a competitive approach to reinforcement learning. [link](https://arxiv.org/abs/1803.07055)

[3] OpenAI Blogpost on ES vs RL. [link](https://openai.com/blog/evolution-strategies/#:~:text=Injecting%20noise%20in%20the%20parameters.&text=However%2C%20RL%20injects%20noise%20in,guess%20and%20check%E2%80%9D%20on%20parameters.)


**Summary Of Recommendation:**

Updating to weak reject after reading the author's responses. Also reducing confidence from 4 to 3 since I'm not very sure if I'm under-assessing the "contextual" part of the paper.

======================Original Content=======================

While the paper is well-written, I cannot recommend acceptance of this paper, given that the major result does not show significant difference from existing methods.

---

> ### Author Response · Authors · 2022-08-23
> **Response to Reviewer Rejd**
>
> We thank the reviewer for their time, their suggestions, and for the opportunity to make some clarifications.
>
> ### Connection to Evolution Strategies
> We first, would like take the time to differentiate our method from algorithms in the realm of Evolution Strategies (ES) as they are conceptually very different and hard to use in our setup.
> ES approaches typically consider learning the DNN policy parameters as the black-box optimization problem and leverage the episode performance for evaluating network configurations.
> We typically have in the order of several hundred to several thousand of these DNN parameters.
> Moreover, ES strategies are hard to use in contextual setups as they require rollouts with different DNN parameters to be comparable. Yet, in the contextual case the performance of the rollout does not only depend on the DNN parameters but also on the context (e.g. the goal) which makes the evaluation much more noisy as a good DNN parameter vector could still achieve a poor performance just because it has been evaluated in a hard context.
> Standard ES approaches such as ARS [1] or ES [2] are completely ignorant to the context and therefore can also not attribute the performance differences to the context.
>
> In contrast, our BBRL algorithmic setting does not perform black-box optimization on the level of a global DNN control policy with several thousand parameters but on the level of local control parameters (in the range of 20-50 dimension).
> Hence, our "black box" is a much lower dimensional problem.
> We do still use a DNN policy $\pi(w|c)$ with a high number of parameters to adapt the local control parameters to the context, yet, the DNN parameters are, as opposed to standard ES, *not* treated as a black box.
> Instead, the DNN policy is updated using policy gradient, which utilizes the context information as well as the derivatives of $\log \pi(w|c)$.
> Hence, our method could probably also be considered as a gray-box approach as it incorporates the context information into the black-box search which is a much more powerful concept for the given tasks than non-contextual ES.
>
> ES approaches that are now able to adapt the parameters based on the context have so far been limited to linear adaptation strategies, such as contextual CMA-ES [3] or contextual MORE [4].
> This is what we referred to as "limited to linear methods".
> These methods do not scale to non-linear mappings, hence, to the best of our knowledge, this is the first work that investigates non-linear contextual black-box optimization.
> Our work also shows why contextual black box optimization is hard for standard deep RL algorithms due to the higher required precision of action selection.
> Here, we showed that principled trust region updates are required as opposed to more heuristically motivated and approximate proximal updates, such as PPO.
> We believe that this insight is also very important for future work on optimizing more abstract action parametrizations such as MPs or other controllers.
>
> While we believe that standard "context-ignorant" ES strategies for optimizing DNN controllers are not very well suited for our setup, we ran some experiments on the Reacher and HopperJump task which confirm our initial assumption of the poor performance.
> We also tested the contextual ES strategy contextual MORE which is limited to a linear context adaptation to adapt the MP parameters similar as in our algorithm.
> While contextual MORE can perform better than ES, its performance is limited by the linear adaptation mechanism.
> We will update our paper in due time when all experiments and seeds are finished.
>
> ### General Benefits of BB approaches
> While it is true that our approach is "only" gaining the benefits other black box approaches have, we believe that the community is not fully aware of the effects of these benefits in the case of standard RL as well as on the disadvantages of dense rewards on the behavior of the learned policy.
> For example, a common believe is that energy efficiency can be achieved by increasing the action costs for dense rewards.
> Yet, our experiments show that dense rewards fall short in expressing energy efficient behavior even with high action costs as the goal costs always dominate.
> Moreover, our method is the first method that transfers the advantages of black-box optimization to the contextual black-box (or rather gray-box, as discussed above) optimization case, allowing the exploitation of these benefits for more complex, goal-conditioned task definitions as well as utilizing parametrized local controllers in the RL loop.

---

> > ### Author Response · Authors · 2022-08-23
> > **Response to Reviewer Rejd**
> >
> > ### References
> > [1] Mania, Horia, Aurelia Guy, and Benjamin Recht. "Simple random search provides a competitive approach to reinforcement learning." In Advances in Neural Information Processing Systems (2018).
> > [2] Salimans, Tim, Jonathan Ho, Xi Chen, Szymon Sidor, and Ilya Sutskever. "Evolution strategies as a scalable alternative to reinforcement learning." arXiv preprint arXiv:1703.03864 (2017).
> > [3] Abdolmaleki, Abbas, Bob Price, Nuno Lau, Luis Paulo Reis, and Gerhard Neumann. "Contextual covariance matrix adaptation evolutionary strategies." International Joint Conferences on Artificial Intelligence Organization (IJCAI), 2017.
> > [4] Tangkaratt, Voot and van Hoof, Herke and Parisi, Simone and Neumann, Gerhard and Peters, Jan and Sugiyama, Masashi. "Policy search with high-dimensional context variables." In Proceedings of the AAAI Conference on Artificial Intelligence (2017).

---

> > ### Comment · Reviewer_Rejd · 2022-08-26
> > **Response to author's comments.**
> >
> > Thanks for the response. I agree that I may have overlooked the "contextual" part of the experiment design and that is a novel proposal of the paper. However, I would like to mention that:
> >
> > 1. Fully black-box methods have been known to handle networks with intermediate number of parameters (on the order of 1-10k), and even for large convolutional neural networks [1]. So I think that context-ignorant black-box methods would still serve as a strong baseline for the paper, and encourage the authors to try further.
> >
> > 2. While the community may not be fully aware of the benefits of BB approaches, demonstrating the known benefits with more specific instances such as energy efficiency may not be sufficient for a conference like CoRL.
> >
> > I would raise my score to "Weak Reject" based on the author's response.
> >
> > [1] Evolving Deep Convolutional Neural Networks for Image Classification. Sun. et al.

---

> > > ### Author Response · Authors · 2022-08-26
> > > **Re: Reviewer Response**
> > >
> > > 1. We now uploaded our revised version where we evaluated against the ES approach from [1] using the implementation from ray [2] on the Reacher (sparse and dense), Meta-World, and HopperJump task. We got a rather poor performance, which is rather unsurprising as context-ignorant ES strategies do not only have to deal with a larger black box (which we agree that black box problems in this range have been solved using ES) but the parameter evaluations are also very "noisy" as the algorithm cannot attribute performance differences to the context (it only attributes it to the parameters). Hence, context ignorant ES has a very hard time optimizing the given objective even though the size of the black-box would be manageable for ES, not just because the size of the black box problem but mainly due to the "noise" added to the evaluations.
> > >
> > >
> > > 2. Making people aware of the benefits of black-box optimization in specific tasks is not the main contribution of the paper, but presenting an algorithm that can deal with the contextual black-box setting and show how powerful this setting is. We believe that the main reason why the contextual black-box setting is not very popular so far is that there was no algorithm that can deal with the accuracy requirements for action selection in this setting. This changes with the given paper, allowing the exploration of new problem formulations for complex robot learning tasks.
> > >
> > >
> > > [1] Salimans, Tim, Jonathan Ho, Xi Chen, Szymon Sidor, and Ilya Sutskever. "Evolution strategies as a scalable alternative to reinforcement learning." arXiv preprint arXiv:1703.03864 (2017).
> > > [2] https://docs.ray.io/en/latest/rllib/rllib-algorithms.html#es

---

### Official Review · Reviewer_FPE5 · 2022-07-26

**Originality:** Very Good
**Technical Quality:** Good
**Clarity Of Presentation:** Very Good
**Impact:** 4

**Recommendation:**

Weak Accept: I recommend accepting the paper, but will not argue for my recommendation if the majority of other reviewers have a different opinion.

**Summary:**

This paper presents a deep episode-based reinforcement learning algorithm by adapting an existing step-based RL algorithm (differentiable trust region layer) to instead select a single action per episode, where the action involves selecting the parameters for a parametrized motion primitive which will be executed for the full episode. The authors then demonstrate that the proposed episode-based RL algorithm can more easily learn from sparse rewards for a number of continuous robotic control tasks in simulation. The main contribution of this work appears to be the first deep contextual episode-based RL algorithm and experimental results suggesting the utility of such an algorithm for robotic tasks.

**Issues:**

(1) Please add comparisons to (a) hierarchical RL algorithms with similarly parameterized primitives or (b) comparisons where the choice of motion primitive is changed (ie. something different from ProMP) to (a) provide a fairer algorithmic comparison to prior approaches and (b) study the impact of the choice of motion primitive on experimental results

(2) The paper would be much stronger if it is evaluated on physical hardware. RL training need not be performed on physical hardware, but transferring a learned policy from simulation to the real world would be beneficial in making a better case for the relevance of the proposed approach to robotic applications.

**Quality Of The Limitations Section:**

Limitations are addressed clearly

**Reviewer Expertise:**

4: The reviewer is confident but not absolutely certain that the evaluation is correct

**Robotics Focus:**

Highly relevant to robotics but no hardware experiments

**Strengths And Weaknesses:**

Strengths:

(1) While the algorithmic innovation with respect to the differentiable trust region layer work appears minimal (all that seems to be required is applying this method for one step and feeding the results to a parameterized motion primitive), the idea to develop a deep episode-based RL algorithm definitely seems to be novel and well motivated for robotic tasks.

(2) The experimental results do clearly indicate that the proposed approach is better able to handle sparse and non-Markovian rewards than step-based RL algorithms and leads to more energy efficient policies as a result across a number of standard simulation environments.

(3) The method and experimental results are very clearly and thoroughly described and easy to understand.

Weaknesses:

(1) My interpretation of this paper was mainly that choosing a more appropriate action space (in this case ProMPs) can lead to more efficient RL. As a result, while comparison to step-based RL algorithms is good, I believe that these comparisons alone are insufficient since they have access to a much less expressive action space than the presented algorithm. To this end, the paper would be significantly strengthened by adding comparisons to hierarchical RL algorithms with similarly parameterized primitives or comparisons where the choice of motion primitive is changed (ie. something different from ProMP).

(2) While I believe that the proposed approach would be useful for practical physical robotic experiments, it would significantly strengthen the paper to evaluate it on physical hardware.

(3) [Minor]: It would be helpful to discuss connections between the proposed approach and ideas from the bandits and contextual bandits community. These papers may be of particular interest: https://arxiv.org/abs/1807.09809, https://arxiv.org/pdf/2206.08921.pdf

**Summary Of Recommendation:**

Overall I am positive on this paper and like the idea of essentially doing 1-step RL with classical motion primitives for robotic learning. The primary weaknesses of the paper are (1) the lack of comparison to hierarchical RL methods, which would be much fairer and interesting than comparison to purely step-based RL approaches and (2) the lack of physical experiments. Otherwise the contribution and experiments  are very clearly and thoroughly presented, so I am learning towards an accept decision at this time.

---

> ### Author Response · Authors · 2022-08-23
> **Response to Reviewer FPE5**
>
> We thank the reviewer for their time and their insights.
> We especially agree with the reviewer’s comment that this work leverages a more expressive action space compared to related step-based policies.
>
> ### Expressive Action Space
> This more complex action space is exactly the point of our method.
> A more expressive action space allows to learn behavior of higher quality, but also requires more precise policy updates (TRPL vs PPO).
> We also want to emphasize that this work aims to show this benefit over common deep RL policies.
> Therefore, we find comparing to common learning approaches like PPO to be important here.
>
> The focus of our work is not on developing or analyzing the effects of using different trajectory generators.
> In contrast, we want to emphasize that the proposed method is independent of the trajectory generator and focuses on showing the benefits of combining trajectory generation and deep RL approaches.
> We have chosen ProMPs here, as they have shown to be easy to use in the past.
>
> Considering hierarchical approaches in the context of episode-based RL is very interesting.
> Related works such as [1-3] have considered learning hierarchical policies in the episode-based RL context.
> Yet, these methods focus on learning versatile skills by optimizing a mixture of experts model and, thus, focus on a different problem setting.
> Additionally, we do not optimize a hierarchical policy here and, thus, did not conduct any comparisons to these algorithms.
> Note that our algorithm could be used to update the single components for the hierarchical approaches as well and, therefore, would potentially benefit from these approaches (they currently use a linear adaptation scheme to adapt the MP parameters to the context).
>
> ### Real World Experiments
> We agree with the reviewer that real world applications would be beneficial to demonstrate the advantages of our method.
> While we cannot show any experiments at this point, movement primitives have shown to have good sim2real capabilities [4].
> This is mainly due to the fact that generated trajectories are smooth and can easily be followed by simple tracking controllers, such as PD-controllers.
> Hence, we firmly believe that this method also provides value for real world applications and consider confirming the sim2real capabilities as future work.
>
> ### Connection to Contextual Bandits
> Indeed, our method is effectively an infinitely armed contextual bandit with a continuous context space.
> However, this scenario is typically not considered in the bandit setting.
> Often action and/or context spaces are discretized as also done in the the two suggested works.
>
> ### References
> [1] Celik, Onur, Dongzhuoran Zhou, Ge Li, Philipp Becker, and Gerhard Neumann. "Specializing Versatile Skill Libraries using Local Mixture of Experts." In Conference on Robot Learning, pp. 1423-1433. PMLR, 2022.
> [2] Daniel, Christian, Gerhard Neumann, and Jan Peters. "Hierarchical relative entropy policy search." In Artificial Intelligence and Statistics, pp. 273-281. PMLR, 2012.
> [3] Bruno Castro Da Silva, George Konidaris, and Andrew G. Barto. "Learning parameterized skills." In International Conference on Machine Learning, pp. 1443–1450. 2012.
> [4] Klink, Pascal, Hany Abdulsamad, Boris Belousov, and Jan Peters. "Self-paced contextual reinforcement learning." In Conference on Robot Learning, pp. 513-529. PMLR, 2020.

---

> > ### Comment · Reviewer_FPE5 · 2022-08-24
> > **Thank You for Your Response**
> >
> > I still disagree that "the proposed method is independent of the trajectory generator". While the RL algorithm itself may not depend on the trajectory generator, the quality of the results surely depends significantly on the choice of trajectory generator used. I also still believe that  RL algorithms with more expressive action spaces (ie. autonomously discovered skills) do still address the same general setting, as these algorithms are simply learning more expressive actions spaces rather than using hand-defined ones (such as ProMPs). I think that while the response doesn't quite address my concerns, my general opinion of the paper remains positive and my rating remains unchanged (Weak Accept).

---

> > > ### Author Response · Authors · 2022-08-25
> > > **Re: Reviewer Response**
> > >
> > > We thank the reviewer for his response and are pleased that the paper continues to be well received.
> > >
> > > We think our previous answer might have been misleading, we apologize for this.
> > > As correctly noted by the reviewer the independence only applies to the algorithm itself.
> > > We agree that the performance can indeed depend on the choice of the trajectory generator, especially if the chosen generator has a largely different degree of complexity, such as basic splines.
> > > Ultimately, we have chosen ProMPs as they have shown to reliably generate good trajectories.
> > >
> > > Regarding the more expressive actions spaces.
> > > While NDP does not autonomously discover skills, it still learns in a more expressive action space than standard step-based methods by leveraging the trajectory generator DMP for action generation.
> > > Maybe this comparison can at least address the reviewer's concerns to some extend.
> > > To incorporate the reviewer's suggestions in our future work, we would kindly ask for some details which algorithms are in their opinion relevant to make this comparison.
> > > We currently assume that the reviewer was referring to methods similar to [1]. It should be noted, however, these methods still rely in one way or another on step-based algorithms for policy optimization. While exploration behavior might be better compared to purely step-based RL, they too have problems on tasks with sparse and non-Markovian rewards.
> > >
> > > [1] Osa, Takayuki, Voot Tangkaratt, and Masashi Sugiyama. "Discovering diverse solutions in deep reinforcement learning by maximizing state–action-based mutual information." Neural Networks 152 (2022): 90-104.

---

### Official Review · Reviewer_3iLp · 2022-07-27

**Originality:** Very Good
**Technical Quality:** Very Good
**Clarity Of Presentation:** Good
**Impact:** 4

**Recommendation:**

Weak Accept: I recommend accepting the paper, but will not argue for my recommendation if the majority of other reviewers have a different opinion.

**Summary:**

The paper proposes a movement primitive based approach to reinforcement learning - instead of learning a function approximator with state conditioning, the policy is parameterized as a contextual movement primitive that translates a parameter vector to an open loop action trajectory to be deployed in the environment.  The advantage of this kind of approach is that it allows for generating smoother trajectories, and has applicability in situations where it is natural to define sparse or non-markovian rewards. Extensive experiments show the benefit of the approach in non-markovian reward settings and its relevance to energy minimization.

**Issues:**

My concerns can be found in the weaknesses section above.

**Quality Of The Limitations Section:**

Limitations are addressed clearly

**Reviewer Expertise:**

2: The reviewer is willing to defend the evaluation, but it is quite likely that the reviewer did not understand central parts of the paper

**Robotics Focus:**

Highly relevant to robotics but no hardware experiments

**Strengths And Weaknesses:**

**Strengths**

1. The paper propose a new method based on trust region optimization for movement primitive parameterized policies that allows for successful learning on manipulation tasks using reinforcement learning. The importance of trust region optimization for policy optimization is shown by an ablation.
2. The method is very relevant to the robotics community where using movement primitives can naturally speed up learning while ensuring more energy efficient and smoother trajectories.
3. The paper contains extensive experiments in simulated domains that are relevant to robotics. The results on meta-world demonstrate convincingly that the method can learn and achieve a higher asymptotic performance than step-wise RL methods which was surprising. The proposed method also allows to incorporate non-markov or sparse rewards while training which can lead to more aligned and energy efficient behaviors.

**Weaknesses:**

1. Paper/Method clarity:
a. In the method section 3.3, it is not clear how the trajectory generator converts a parameter to a state-based trajectory. Since it forms a integral part of their method it might be important to clarify as a background or in appendix.
b. Does the inference of state-based trajectory use any dynamics or inverse-dynamics information? In that case the applicability of the method is quite limited and the baselines compared should incorporate dynamics knowledge.
c. How is the PD controller tuned? Are the samples used for tuning PD controller counted in the sample efficiency plots?
2. Baselines:
a. Efficient off-policy baselines like SAC are missing for a fair comparison in the experiments for domains like meta-world.

    b.  Can the authors give an intutive understanding of why PPO gets stuck in local optima for the meta-world tasks?

3. Engineering movement-primitives:
a. How does the movement primitive scale to high-dimensional or long horizon environments? When the state space is large the importance ratio used in PPO may suffer from large variance because unlike step-based PPO now we have reweighting of large dimensional states?

    b. How is the movement primitive decided? The optimal policy’s state visitation should lie in the space of valid movement primitives and cannot be decided without oracle information and is hard to approximate in tasks like locomotion?

**Summary Of Recommendation:**

The paper is very relevant to the robotics community and presents a method to incorporate movement primitives in the RL framework. The paper has extensive experiments which makes the method convincing but the method writing needs more work and clarity. The ability to incorporate structural prior in policy in order to address sparse reward and non-markovian tasks in nicely addressed in this work.

---

> ### Author Response · Authors · 2022-08-23
> **Response to Reviewer 3iLp**
>
> We thank the reviewer for their time and appreciate the kind words for our method. We will address concerns and questions raised by the reviewer below.
>
> ### Trajectory Generation
> First, while using ProMPs for trajectory generation is one option, other movement primitives, splines, or much more complex and specialized controllers could be used instead.
> Ultimately, ProMPs are just a choice that have shown to work well and reliably generate good trajectories.
> Hence, we do not see it as the scope of this paper to go into the details of ProMPs and will only provide an overview here.
>
> ProMPs represent a distribution over trajectories parameterized through a linear relationship in a weight space
>
> $$
> \tau = \Phi_{0:T}^\intercal w + \epsilon_{y},
> $$
>
> where $\Phi_{0\:T}$ are the radial basis function features for each degree-of-freedom over the time horizon $T$, $w$ are the weights, and $\tau$ is a whole trajectory.
> A distribution over trajectories ${\tau}$ can be inferred from data
>
> $$
> p(\tau ; \mu_w, \Sigma_w) = \mathcal{N}(\tau \~|\~ \Phi_{0:T}^\intercal \mu_w, \Phi_{0:T}^\intercal \Sigma_w \Phi_{0:T}\~+ \sigma_y^2 I),
> $$
>
> where the parameters $w$ define the mean $\mu_w$ and the covariance $\Sigma_w$.
> However, this formulation is more likely to be found in domains other than RL, such as imitation learning.
> For RL, we also consider the linear model from the first equation, but with $\epsilon_{y}=0$ and the weights ${w}$ are now determined/sampled through our policy/search distribution $\pi_\theta({w}|{c})$, which introduces a non-linear dependency to the context ${c}$.
> This relationship is also observable in Figure 1 in the paper.
>
> ### Trajectory Assumptions
> Our general approach does not make any assumptions regarding the dynamics or the inverse-dynamics.
> As stated above, we do not even make any assumptions how the trajectory generator is converting its parametrization to the control actions.
> That being said, if additional information, such as dynamics or inverse dynamics, happen to be available and leveraging those would be beneficial, it is possible to do so in the trajectory generator and/or tracking controller.
> For the specific trajectory generation approach with ProMPs we used in our experiments, this information is not required.
>
> ### Tracking Controller
> The P- and D-gains of the PD-controller are rather trivial and typically set to $1.0$ and $0.1$, respectively, for all joints.
> The values have been chosen empirically based on a handful of random trajectory samples.
> They are not included in the plots, but due to their low number they would not have altered the overall results.
> The actual choice of P- and D-gains in this setting is in practice rather uninteresting as the learned policy can adjust its outputs to yield the desired trajectory.
>
> ### Baselines
> We agree that SAC is a strong off-policy baseline we have not considered in our experiments.
> We are currently in the process of evaluating it on some of the tasks.
> First results indicate that SAC also has the same issues as other step-based algorithms.
>
> When it comes to the comparison with step-based learning on Meta-World.
> Our main goal was to demonstrate that episode-based methods can achieve a competitive asymptotic performance even in tasks heavily designed for step-based methods.
> Admittedly, this comes with the cost of an increased sample complexity, yet, unlike step-based methods, episode-based methods also work well in other domains without well shaped dense rewards.
> When directly comparing PPO with BBRL-TRPL, we found PPO's heuristic approach for trust regions works well on average, however it has its limitations leading to premature convergence.
> This is also evident when compared to the performance of BBRL-PPO, which performs significantly worse.
> Properly enforced trust regions are critical to working in the BBRL setting to provide stability.

---

> > ### Author Response · Authors · 2022-08-23
> > **Response to Reviewer 3iLp**
> >
> > ### Scalability
> >
> > Scalability to long horizon tasks is straightforwardly possible as the ProMP's radial basis features can directly be adjusted to different trajectory lengths.
> > This also does not alter the parameter space for both the policy or the ProMP, i.e. learning does not inherently become more complex.
> > One limitation here is that we can only consider finite horizon tasks, however with arbitrary length.
> > Adding degrees-of-freedom, on the other hand, changes the number of parameters of the ProMP depending the employed number of basis functions.
> > While we think this will scale well for the application areas we consider (continuous robotic control with finite horizons), really high dimensional problems with complex behaviors, such as the humanoid tasks, might be challenging to scale.
> > Different trajectory generators and modifications, such as skill sequencing or replanning, could support this, however this is not our main focus for this work.
> >
> >  > "When the state space is large the importance ratio used in PPO may suffer from large variance because unlike step-based PPO now we have reweighting of large dimensional states?"
> >
> > We believe the reviewer meant the action space in this question as the context space of the BBRL setting is a subset of the observation space and normally much smaller.
> > When it comes to the action space, the increased dimensionality can indeed lead to larger variances.
> > This is the main reason we use TRPL for learning the policy, as its trust regions are able to limit how much this variance can effect the overall update steps.
> > As shown in the paper, the importance ratio clipping of PPO is not sufficiently able to constrain the policy in the BBRL setting and performs worse than BBRL-TRPL.
> >
> > ### Choice of Trajectory Generator
> > Indeed specifying an expressive trajectory generator is crucial for successfully solving a task, however, there is only one hyper-parameter to choose when using ProMPs, i.e., the number of basis functions.
> > We observed that choosing this hyper-parameter is relatively simple, similar to choosing the number of layers or neurons for a DNN policy.
> > One case, where our current choice of trajectory generator might have problems is for highly complex locomotion tasks.
> > In this case, we would need cyclic basis functions which exist for most MP formulations.
> > Moreover, future work can focus on infinite horizon tasks by considering sequencing sub-trajectories and replanning strategies.
> > Here, we could sequence basic movements to complex skills which would simplify choosing the correct trajectory generator.
> > We, therefore, consider this line of work as an important extension and future work.

---

> ### Author Response · Authors · 2022-08-27
> **Revised Version**
>
> We have now also included a comparison with SAC in the revised version and hope that this addresses the reviewer's concerns. Details can be found in the global comment above.

---

### Official Review · Reviewer_xpuk · 2022-07-29

**Originality:** Fair
**Technical Quality:** Fair
**Clarity Of Presentation:** Good
**Impact:** 2

**Recommendation:**

Weak Reject: I recommend rejecting the paper, but will not argue for my recommendation if the majority of other reviewers have a different opinion.

**Summary:**

This paper presents an experimental evaluation of episode-based RL (ERL) on a series of simulated tasks, starting from a reacher, box pushing, ball in a cup, to table tennis. The paper starts by proposing a trust-region based ERL algorithm, and evaluates it against step-based approaches on setting with sparse reward functions, and show it performs better than PPO, even with dense rewards.

**Issues:**

1. It would be good to add an additional baseline of TRPL in the comparisons.
2. Authors should improve the quality of figures and experimental section
3. Overall technical contributions are somewhat minor.

**Quality Of The Limitations Section:**

Limitations are addressed clearly

**Reviewer Expertise:**

4: The reviewer is confident but not absolutely certain that the evaluation is correct

**Robotics Focus:**

Relevant but unlikely to deploy to hardware in near future

**Strengths And Weaknesses:**

Strengths:
1. Well motivated problem setting
2. A large range of simulated environments used in the evaluation

Weaknesses:
1. The contributions are minor beyond experimental evaluation in simulation. The paper reformulates and existing approach for an episodic setting, and evaluates it using a motion primitive algorithm.
2. Since all evaluation is in sim, and most experiments take the order of 10M steps, real-world transfer of the approach is hard to determine.
3. The experiments are somewhat hard to understand. The states and action spaces are not well-defined, the cost functions are not clear, and the plots are extremely hard to read. For example Figurers 2,3 don't have titles or legends which makes it hard to understand the results. I wrote my own legends on the figures to make sense of the analysis. I would recommend authors to update these figures to make them easier to follow.
4. Why do authors compare against PPO, but not TRPL? Since PPO does better than BBRL-PPO, I would suspect that TRPL might do better than BBRL-TRPL. Authors should add that comparison, and use it as oracle with dense rewards maybe?


**Summary Of Recommendation:**

The paper's contributions are minor, and the applicability to real robots is questionable due to large number of samples needed. The paper has some really interesting simulation environments though, and I appreciate such a thorough experimentation with competitive baselines.

---

> ### Author Response · Authors · 2022-08-23
> **Response to Reviewer xpuk**
>
> We thank the reviewer for their time and valuable feedback to improve our submission.  Below, we paraphrase and address concerns and questions raised by the reviewer.
>
> ### Novelty
> While we agree that the pure algorithmic contributions are limited, we still think our work presents a significantly novel idea and supports this with a strong experimental evaluation.
> This work effectively combines the advances from more "traditional" trajectory planning with those of deep RL.
> To the best of our knowledge, highly non-linear policies with trust region updates were not applied to the episode-based RL case so far.
> Moreover, our work is novel in that sense that it shows how to leverage non-linear contextual black-box optimization for learning movement primitives.
> Generally, this setting allows the use of locally valid controllers/trajectory generators in much more complex applications as shown in our experiments.
> Existing approaches are limited to using linear models [1,2] and are not able to model complex non-linear relationships between context and movement primitive parametrization.
> Additionally, we provide a thorough discussion on step-based vs episode-based RL and showcase the advantages of both methods in the presence of different scenarios such as dense, sparse, and non-Markovian reward.
> Some benefits which we addressed in our work are also mentioned in [3], but we are not aware of a detailed discussion of such advantages in the literature.
> Lastly, we demonstrate that sparse and non-Markovian rewards can have significant advantages to obtain high-quality behaviors in comparison to the commonly used dense rewards.
> For example, we showcase that skills learned in the episode-based setting tend to be more energy-efficient.
> While it is a common believe that energy efficiency can be achieved by varying the action costs for step-based methods, our analysis shows that these methods fail for high action costs and hence, can not produce energy efficient motions.
>
> ### Sim2real Transfer
>  We agree with the reviewer that a direct application of this method to real world environments is hard to determine.
>  However, movement primitives have been shown to be directly applicable for sim2real transfer, e.g. [4], if good tracking controllers on the real robot are available.
>  Consequently, for real world applications, we would first train in simulation and then adapt the policy to the real world setting if necessary.
>  Moreover, we currently consider a random initial policy.
> To improve the sample complexity we could warm start the initial parametrization by leveraging expert trajectories, which could be considered common practice with movement primitives.
>
> ### Clarity of Experiments
> Thank you for the suggestions, we now updated the detailed descriptions in Appendix B for all the custom tasks and provided some more description in the main text in our revised version which we will upload in due time when all additional experiments are finished.
> Overall, the state and action spaces as well as the dense rewards are kept consistent with what is being used in well known benchmarks, such as OpenAI gym or Deepmind control suite.
> For sparse rewards we typically consider time-sparse rewards, i.e. the task reward is only returned in the last time step.
> The Meta-World tasks stay the same as originally defined.
> The context space is always represented as a subset of the observation space including only the stochastic elements of the first time step, e.g. randomized goal or object positions.
> All other parts of the observation space that are constant in the first time step, due to an identical initialization, can be ignored for the context space in the BBRL setting as they do not provide any value to the trajectory adaptation.
> Lastly, we are planning to open source the environments alongside the camera-ready version.
>
> Regarding the figures, we would like to ask for some clarification what exactly is hard to understand and should be improved.
> We tried to avoid separate legends for each sub-figure to avoid clutter and redundancy, hence each figure has only one legend that is applicable for all sub-figures.
> To this end, we kept colors consistent throughout all plots, i.e. the same algorithm has the same color everywhere.
> The content of sub-figures as well as special cases, such as different lines, are described in the caption.
> For analyzing our results, we reference the specific sub-figures in the text.
>
> ### TRPL Comparison
> We agree with the reviewer that TRPL is a valuable baseline and we are in the process of adding it to our experiments.
> As expected, our initial experiments show a similar or improved performance of TRPL over PPO for the dense rewards but TRPL still performs worse for the remaining reward settings.
> We plan to publish the results in a few days when all seeds are available.

---

> > ### Author Response · Authors · 2022-08-23
> > **Response to Reviewer xpuk**
> >
> > ### References
> > [1] Tangkaratt, Voot and van Hoof, Herke and Parisi, Simone and Neumann, Gerhard and Peters, Jan and Sugiyama, Masashi. "Policy search with high-dimensional context variables." In Proceedings of the AAAI Conference on Artificial Intelligence (2017).
> > [2] Abdolmaleki, Abbas, Bob Price, Nuno Lau, Luis Paulo Reis, and Gerhard Neumann. "Contextual covariance matrix adaptation evolutionary strategies." International Joint Conferences on Artificial Intelligence Organization (IJCAI), 2017.
> > [3] Salimans, Tim, Jonathan Ho, Xi Chen, Szymon Sidor, and Ilya Sutskever. "Evolution strategies as a scalable alternative to reinforcement learning." arXiv preprint arXiv:1703.03864 (2017).
> > [4] Klink, Pascal, Hany Abdulsamad, Boris Belousov, and Jan Peters. "Self-paced contextual reinforcement learning." In Conference on Robot Learning, pp. 513-529. PMLR, 2020.

---

> > > ### Comment · Reviewer_xpuk · 2022-08-28
> > > **response to author comments**
> > >
> > > Thanks for the additional TRPL comparison, which completes the comparison experiments in the paper. Also thank you for the additional experiments. I still think that the novelty of the paper is minor, though I agree that the simulation experimentation is good. I also find the lack of direct real-world transfer a little concerning. Overall, I am willing to change my review to a weak accept, but I encourage the authors to discuss the sim-to-real limitations and their proposed approach for the same in the final paper.

---

> > ### Author Response · Authors · 2022-08-26
> > **Revised Version**
> >
> > We have now improved the figures and added step-based TRPL as baseline in the revised version and hope we addressed the reviewers concerns with that. If the reviewer has additional suggestions in mind, we are happy to include them.
> > For further details please see global comment above.

---

### Author Response · Authors · 2022-08-26
**Revised Version**

We again thank all reviewers for their valuable feedback and believe the paper has now a higher quality than before.
Currently, the main paper has temporarily 8.5 pages, as allowed by the Program Chairs for the rebuttal, we will fix this for the camera-ready version.

Below is a summary of the conducted changes and new findings.
All changes in the text are highlighted in red.

**Changelog:**

[27.08.2022]
- Added SAC baseline for Reacher, Box Pushing, and Hopper Jump
- Added OpenAI's ES baseline (full black-box approach with NNs) for Box Pushing

[26.08.2022]
- Added contextual MORE baseline (linear contextual adaptation) for Reacher, HopperJump, and Beer Pong
- Added OpenAI's ES baseline (full black-box approach with NNs) for Reacher, Meta-World, and HopperJump
- Added step-based TRPL baseline for Reacher, Box Pushing, Meta-World, HopperJump, and Table Tennis
- Added discussion and comparison with full black-box approaches
- Improved description of custom tasks
- Improve figure legends and added titles
- Minor fixes

The SAC baseline is currently still running and will be added tomorrow.

**General observations:**
- As expected, step-based TRPL performs better on dense rewards, however, our approach is still competitive and achieves high quality solutions. Yet, similar as the other step-based methods, TRPL does not perform well on sparse and non-Markovian rewards.
- The full black-box approach ES is not able to achieve a competitive performance on any of the environments due to the inability to cope with noisy evaluations.
- The linear adaptation method CMORE is limited by its model capacity and only able to solve the tasks partially.

---

### Meta-Review · Area_Chair_pa2s · 2022-08-15

**Recommendation:** Accept (Poster)
**Confidence:** 4

**Metareview:**

Scores: xpuk: Weak Accept, 3iLp: Weak Accept, FPE5: Weak Accept, Rejd: Weak Reject

Quality: The paper is of high quality providing new theory and extensive simulation experiments.

Clarity: The clarity of the paper has been improved in the revision. The additional baselines provide further support for the model.

Originality: The proposed algorithm is interesting and novel.

Significance: The paper provides a solid contribution to robot learning by introducing a stable deep episode-based RL algorithm.

Pros: The paper provides
- a novel episode-based RL algorithm which incorporates movement primitives in the RL framework.
- extensive simulations and baseline comparisons showing robust performance better or comparable to SoTAs.
- the ability to incorporate structural prior in policy and the capability to address sparse reward and non-markovian tasks.

Cons: The major weaknesses of the paper are
-  the lack of physical experiments
-  a sim-2-real assessment
-  a lack of comparisons to other RL algorithms with more expressive action spaces than simply atomic ones.

**Best Paper Nomination:**

No